# Spatial and temporal variability of $p$CO$_2$ and CO$_2$ emissions from the Dongjiang River in South China

Boyi Liu[1], Mingyang Tian[2], Kaimin Shih[3], Chun Ngai Chan[1], Xiankun Yang[4], Lishan Ran[1,*]

5   [1]Department of Geography, the University of Hong Kong, Hong Kong SAR, China
[2]Institute for Geology, Center for Earth System Research and Sustainability (CEN), Universität Hamburg, Hamburg, Germany
[3]Department of Civil Engineering, the University of Hong Kong, Hong Kong SAR, China
[4]School of Geographical Sciences, Guangzhou University, Guangzhou,510006, China

[*] *Correspondence to*: Lishan Ran (lsran@hku.hk)

**Abstract.** CO$_2$ efflux at the water–air interface is an essential component of the riverine carbon cycle. However, the lack of spatially resolved CO$_2$ emission measurements prohibits reliable estimation of the global riverine CO$_2$ emissions. By deploying floating chambers, seasonal changes in river water CO$_2$ 15  partial pressure ($p$CO$_2$) and CO$_2$ emissions from the Dongjiang River in South China were investigated. Spatial and temporal patterns of $p$CO$_2$ were mainly affected by terrestrial carbon inputs (i.e., organic and inorganic carbon) and in-stream metabolism, both of which varied due to different land cover, catchment topography, and seasonality of precipitation and temperature. Temperature-normalized gas transfer velocity ($k_{600}$) in small rivers were 8.29 ± 11.29 m d$^{-1}$ and 4.90 ± 3.82 m d$^{-1}$ for the wet season and dry 20  season, respectively, which were nearly 70 % higher than that of large rivers (3.90 ± 5.55 m d$^{-1}$ during the wet season and 2.25 ± 1.61 m d$^{-1}$ during the dry season). A significant correlation was observed between $k_{600}$ and flow velocity but not wind speed regardless of river size. Most of the surveyed rivers were net CO$_2$ source while exhibiting substantial seasonal variations. The mean CO$_2$ flux was 300.1 and 264.2 mmol m$^{-2}$ d$^{-1}$ during the wet season for large and small rivers, respectively, 2-fold larger than that 25  during the dry season. However, no significant difference in CO$_2$ flux was observed between small and large rivers. The absence of commonly observed higher CO$_2$ fluxes in small rivers could be associated with the depletion effect caused by abundant and consistent precipitation in this subtropical monsoon catchment.

## 1 Introduction

River networks act as a processor that transfers and emits the carbon entering the water, rather than just a passive pipe that transports carbon from the terrestrial ecosystem to the ocean (Cole et al., 2007; Battin et al., 2009; Drake et al., 2018). $CO_2$ emissions at the water–air interface are an essential component of the riverine carbon cycle. $CO_2$ emitted from inland waters to the atmosphere reaches up to 2.9 Pg C $yr^{-1}$, surpassing that transported from land to ocean through rivers (Sawakuchi et al., 2017; Drake et al., 2018).

Understanding the role that rivers play in the global carbon cycle is still hindered by uncertainty on the flux estimate of $CO_2$ emissions from rivers (Cole et al., 2007; Raymond et al., 2013; Sawakuchi et al., 2017; Drake et al., 2018). Riverine carbon emissions have significant temporal and spatial variations, making it challenging to accurately quantify carbon emissions. In addition, watershed geomorphology, hydrological conditions, climate, and other environmental factors can affect the $CO_2$ efflux in rivers (Alin

et al., 2011; Abril et al., 2014; Almeida et al., 2017; Ran et al., 2017a; Borges et al., 2018). Thus, there are substantial differences in $CO_2$ efflux among rivers in different climate regions, or the same river but between different seasons (Denfeld et al., 2013; Rasera et al., 2013). An enhanced understanding of the temporal and spatial characteristics of the water–air $CO_2$ flux will facilitate a more robust estimate. However, global riverine $CO_2$ emission estimates were largely based on data disproportionately focusing

on temperate and boreal regions, including North America and Europe (Raymond et al., 2013; Lauerwald et al., 2015; Drake et al., 2018). More studies are required in other data-poor regions to achieve a more accurate estimate.

    Rivers in tropical and subtropical regions of East Asia and Southeast Asia are among those underrepresented regions that need more attention since they are essential participants in riverine carbon

transport (Ran et al., 2015; Ran et al., 2017b; Drake et al., 2018). The high temperature in this region facilitates a high net primary productivity in the terrestrial ecosystem and intense biochemical activities; both contribute to the carbon input dynamic from soil to rivers (Li et al., 2018). Meanwhile, rivers in this region are under the heavy influence of monsoon climate, and riverine $CO_2$ emissions vary significantly among seasons due to the changes in temperature and precipitation. In addition, different rivers in this

region may have contrasting trends in $CO_2$ dynamic due to different underlying controlling factors. Some rivers have the highest $CO_2$ efflux in the wet season (Li et al., 2013; Le et al., 2018; Ni et al., 2019), while others have the highest $CO_2$ efflux in the dry season (Luo et al., 2019), suggesting that an increase in the wet season runoff can have two distinct consequences. On one hand, recent studies have indicated

that the increased runoff could enhance external carbon inputs and thus $CO_2$ emissions (Hope et al., 2004; Johnson et al., 2008). On the other hand, the increased runoff may result in a dilution of the dissolved $CO_2$ in rivers and accordingly a reduction in $CO_2$ emissions (Ran et al., 2017b; Li et al., 2018). Therefore, it is important to investigate the underlying processes that determine the diverse responses of $CO_2$ emissions to the monsoon climate.

The Dongjiang River (DJR), located in the subtropical South China, is one of the three tributaries of the Pearl River. Previous studies on riverine carbon transport and emissions in the Pearl River system mainly focused on the Xijiang River, which is characterized by widely distributed carbonate rocks, and the estuary area of the Pearl River Delta (Yao et al., 2007; Zhang et al., 2015; Zhang et al., 2019; Liang et al., 2020). Although some studies on chemical weathering and dissolved inorganic carbon transport in the Dongjiang River basin (DJRB) have been conducted (Tao et al., 2011; Fu et al., 2014), there is still a lack of understanding of the characteristics of catchment-wide $CO_2$ emissions from the DJRB. Furthermore, a predominantly hilly landscape combined with abundant precipitation favours the formation of a great number of small rivers in the DJRB (Ding et al., 2015). However, current estimates of basin-wide $CO_2$ emissions from the river network are mostly based on the data from large rivers, and small rivers are heavily underrepresented (Raymond et al., 2013; Drake et al., 2018). Because the controlling factors and the input of carbon could be significantly different between large and small rivers (Johnson et al., 2008; Dinsmore et al., 2013; Hotchkiss et al., 2015; Marx et al., 2017), a more comprehensive quantification of $CO_2$ emissions from small headwater streams is necessary. Therefore, studies on the characteristics of riverine $CO_2$ emissions from the DJRB should be conducted among river size spectrums, and the impact of monsoon needs to be considered.

By using directly measured river water $CO_2$ partial pressure ($pCO_2$) and $CO_2$ emission data from the DJRB and in conjunction with hydrological and physicochemical data, the objectives of this study were to 1) investigate the spatial and temporal pattern of $pCO_2$ and $CO_2$ emissions along stream size spectrum and 2) examine the differences in hydrological and physicochemical controls on $pCO_2$ and $CO_2$ emissions between small headwater streams and large rivers. The results of this study will shed light on the underlying controls of the spatial and temporal distribution of riverine $pCO_2$ and support a refined estimate of regional and global carbon budgets.

## 2 Material and methods

### 2.1 Site Description

The DJR in South China is one of the three major tributaries of the Pearl River system (Figure 1). It has a 562 km long mainstem channel and a drainage area of 35,340 $km^2$ (Chen et al., 2011). Due to its subtropical monsoon climate, precipitation in the DJRB exhibits significant seasonal variability (Figure 2a). The multi-annual average precipitation is about 1800 mm, 80 % of which is concentrated during the wet season from April to September. The Boluo Hydrological Gauge is the lowermost gauge of the Dongjiang River mainstem channel, controlling a drainage area of ~23,000 $km^2$. The multi-annual average water discharge at Boluo Hydrological Gauge is 23.7 $km^3$ (Zhang et al., 2008). About 80–90 % of the discharge is transported during the wet season (Figure 2b). The landscape is characterized by plains and hills, accounting for 87.3 % of the river basin area (Ding et al., 2015), and the dominant land use of the catchment is highly diverse evergreen forests of broad-leaved and needle-leaved species (Ran et al., 2012; Chen et al., 2013). The impacts of human activities on land use vary among three regions in the DJRB. Urban expansion and agricultural activities have substantially altered the land use in the Lower and Middle Dongjiang River Basin (LDJRB and MDJRB), respectively, while the Upper Dongjiang River Basin (UDJRB) is less affected by human activities (Figure 1).

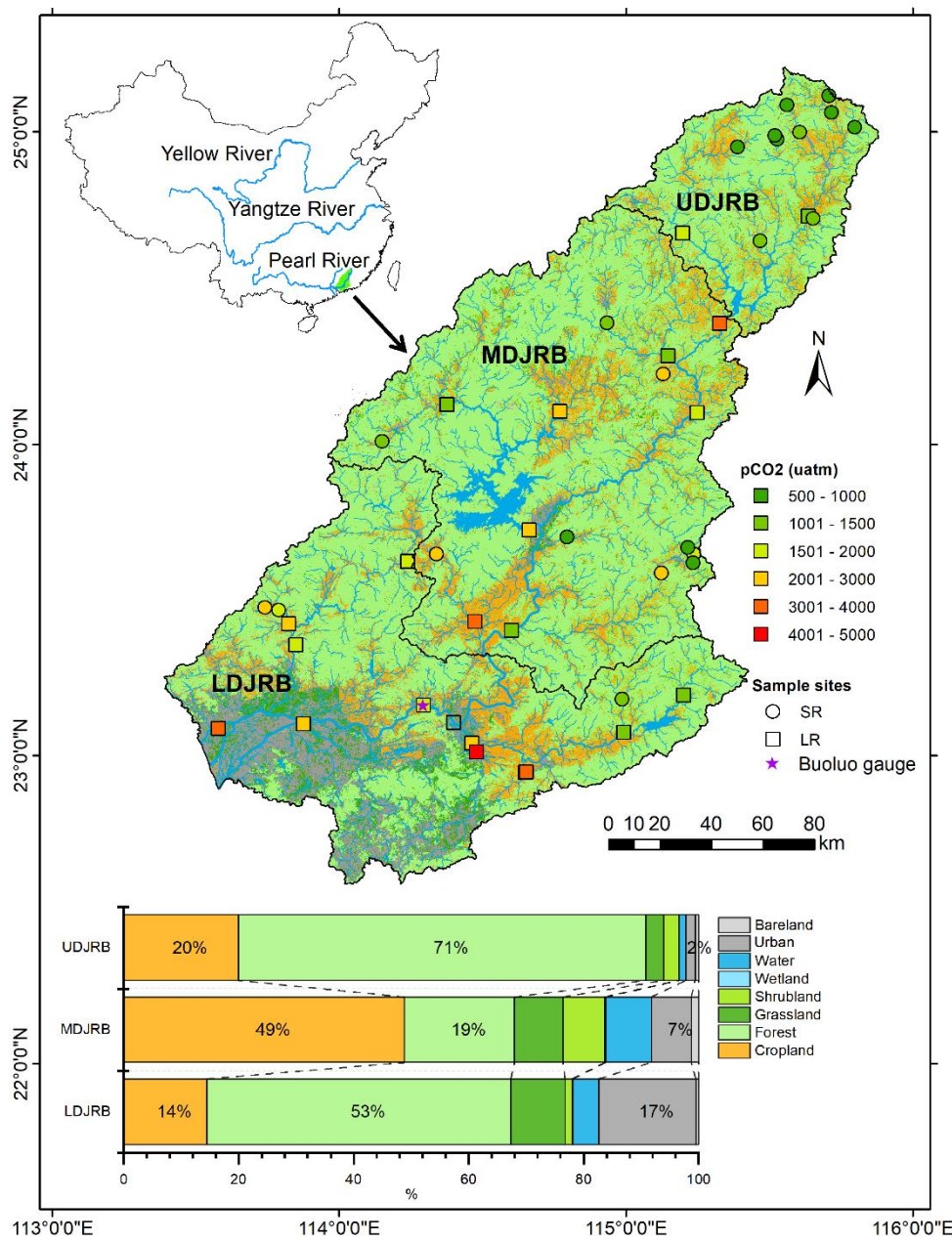

**Figure 1** Sample sites and land cover in the DJRB. Yearly average $p$CO$_2$ at each sample site was displayed. Based on land cover dataset: FROM-GLC10 (http://data.ess.tsinghua.edu.cn).

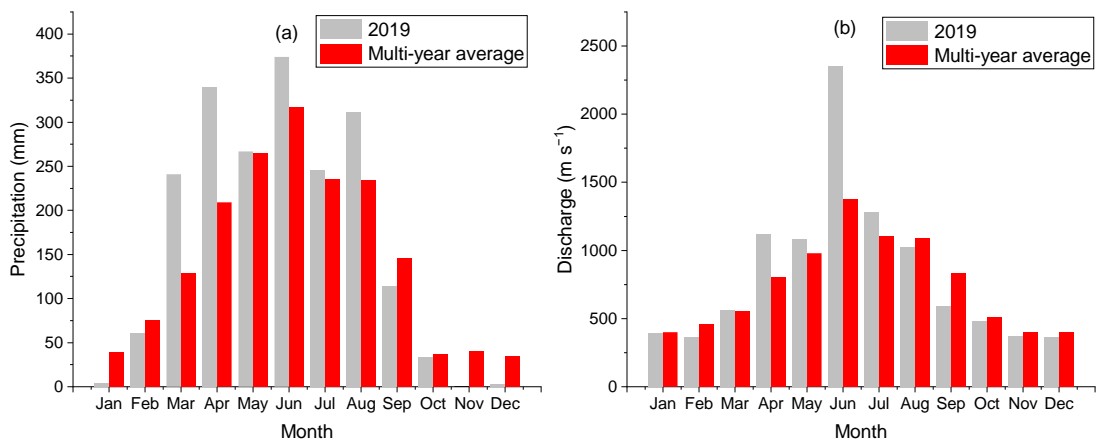

**Figure 2** Monthly variations in (a) precipitation of the DJRB and (b) water discharge at the Boluo hydrological gauge, based on data provided by the Hydrological Bureau of Guangdong Province.

### 2.2 Field Measurements and Analyses

In total, there were 43 sampling sites spanning seven Strahler stream orders. Fourth to seven order streams were mainstem and major tributaries, while first to third order streams were small tributaries. River widths were measured by a laser rangefinder. Sampled rivers were categorized, according to their stream orders, into small rivers (first to third order streams, SR) and large rivers (fourth to seventh order streams, LR). The small rivers had an average width of $15.4 \pm 10.2$ m, while large rivers have an average

width of $180.3 \pm 159.3$ m (Table S1). Those sampling sites were widely distributed in the mainstem and nine major sub-catchments among the three regions with different topographic features and land cover (Figure 1). In order to investigate $CO_2$ emissions during different hydrological conditions, we performed five fieldwork campaigns from December 2018 to October 2019, including three in the wet season (early wet season - late April, middle wet season - early July, and late wet season - late August) and two in the

dry season (middle dry season - December 2018 to early January 2019 and early dry season - late October 2019. Sample sites were measured in the daytime over two weeks for each field trip. Three campaigns in the wet season allowed each sample site to be measured under different hydrological conditions. As for the dry season, the hydrological condition was relatively stable due to low precipitation. However, field measurements conducted during the daytime could lead to an underestimate in $p$$CO_2$ and $CO_2$

emissions (Reiman and Xu, 2019a). Nocturnal $CO_2$ emission rates in rivers could be 27% greater than the daytime rates (Gómez-Gener et al., 2021). During the field trips, water temperature, pH, and dissolved

oxygen (DO) were measured with a portable multiparameter probe (Multi 3430, WTW GmbH, Germany). The pH probe was calibrated before each field trip with standard pH buffers (4.01 and 7.00). Measurements were conducted 10 cm below the water surface.     To evaluate the contribution of metabolism on DO changes, $\Delta CO_2$ and $\Delta O_2$ were calculated as described by Stets et al. (2017) using:

$$\Delta CO_2 = CO_{2w} - CO_{2a} \tag{1}$$

and

$$\Delta O_2 = O_{2w} - O_{2a} \tag{2}$$

Where, $CO_{2w}$ and $O_{2w}$ are measured concentrations of $CO_2$ and $O_2$ in water sample, while $CO_{2a}$ and $O_{2a}$ are the equilibrium $CO_2$ and $O_2$ concentrations ($\mu mol\ L^{-1}$).

Flow velocity was determined by using a Global Water Flow Probe FP111 with a precision of 0.1 m s$^{-1}$, while wind speed at 1.5 m above the water surface was measured with a Kestrel 2500 handheld anemometer and normalized to a height of 10 m (U10) using the equation from Alin et al. (2011). As the flow velocity was measured near the riverbanks, an underestimation of the flow velocity is possible. Flow velocity measured near the riverbanks is only about 40% of the maximum flow velocity at the cross-section (Moramarco et al., 2004; Le Coz et al., 2008). We also collected water for analyzing total alkalinity (TA) and dissolved organic carbon (DOC). Firstly, 100 ml of water samples were filtered through a pre-combusted glass fiber filter (pore size: 0.47 µm, Whatman GF/F, GE Healthcare Life Sciences, USA). Then, 50 ml of water used for TA analysis was titrated with 0.1 mol L$^{-1}$ HCl on the same day of sampling. The remaining 50 ml of water for DOC analysis was poisoned with concentrated $H_2SO_4$ to pH < 2 and preserved in a cooler with ice bags before analysis. DOC was determined by the high-temperature combustion method using a TOC Analyzer (Elementar Analysensysteme GmbH, Langenselbold, Germany) that has a precision better than 3 %.

**2.3 Calculation of $p CO_2$ and $CO_2$ emission flux**

The surface water $p CO_2$ was determined using the headspace equilibrium method, which could avoid the possible overestimation of using TA and pH to calculate $p CO_2$ in rivers with a relatively low pH (Abril et al., 2015).  We used a 625 mL reagent bottle to collect 400 mL of water from ~10 cm below the surface, leaving 225 mL of space filled with ambient air as headspace. The bottle was then immediately capped and shaken vigorously for at least 1 min to achieve an equilibrium between the water and the $CO_2$ in the

headspace(Hope et al., 1994). Then, the bottle was connected to the calibrated Li-850 $CO_2$/$H_2O$ gas analyzer (Li-Cor, Inc, USA), and the equilibrated gas in this closed loop was measured. The measurements at each site were repeated twice, and the average was then calculated. The variation between the two measurements was less than 5%, and the accuracy of Li-850 is within 1.5% of the reading. The ambient air $pCO_2$ ($pCO_2^{air}$) was measured before the headspace measurements and the

chamber deployments. The $pCO_2^{air}$ value varied between 380 and 450 μatm. The original surface water $pCO_2$ ($pCO_2^{water,i}$) was finally calculated by using solubility constants ($K_0$) for $CO_2$ from Weiss (1974), Carbonate constants ($K_1$, $K_2$) from (Millero et al., 2006), and the volume of the flask, headspace, and residual system (line and gas analyzer) (Dickson et al., 2007; Ran et al., 2017a; Tian et al., 2019) using:

$$pCO_2^{water,i} = pCO_2^{headspace,f} + (\frac{Vh+Vr}{Vw})(pCO_2^{h+r} - pCO_2^{headspace,i})/[RTK_0(1 + \frac{K_1}{[H^+]} + \frac{K_1K_2}{[H^+]^2})] \qquad (3)$$

Where, $V_h$, $V_r$ and $V_w$, are the headspace volume, residence system volume, and water volume, respectively. R is the universal gas constant (8.314 J mol$^{-1}$ K$^{-1}$), T is the water temperature in Kelvin (K), and [H$^+$] is the concentration of hydrogen ion. $pCO_2^{heasdspace,i}$ and $pCO_2^{headspace,f}$ are $pCO_2$ before and after the headspace equilibration, respectively. $pCO_2^{h+r}$ is the $pCO_2$ of the mixed gas in the headspace and residual system during the measurement. the $pCO_2^{headspace,i}$ was taken as the $pCO_2$ in ambient air

before the measurement, while $pCO_2^{headspace,f}$ was calculated using:

$$pCO_2^{headspace,f} = pCO_2^{h+r} + (\frac{V_r}{V_h})(pCO_2^{h+r} - pCO_2^{headspace,i}) \qquad (4)$$

To measure $V_r$, we filled the headspace with ambient air, which had a known $pCO_2$, and measured the $pCO_2$ in the closed loop. $V_r$ was then estimated according to equation (3). A comparative analysis of the syringe and bottle headspace method has been conducted to evaluate the accuracy of the headspace

extraction method used in this study (Table S2 and Figure S2). Overall, our method could cause a 1–5% underestimation in $pCO_2$.

To reduce the artificial turbulence induced by anchored chambers, we used a small unmanned boat in the measurement, which allowed us to deploy drifting chambers freely in rivers deeper than 0.2 m and with a high flow velocity up to 2 m s$^{-1}$. During the deployment, $CO_2$ emissions were determined using a

circular, 8.5 L floating chamber with a water surface area of 0.113 m$^2$. The chamber walls were lowered about 2 cm into the water and mounted with a pneumatic rubber tire. The chamber was connected to an

infrared Li-850 $CO_2$/$H_2O$ gas analyzer (Li-Cor, Inc, USA) in a floating storage box through Polyurethane tubes for $CO_2$ analysis. An unmanned boat connected to both the chamber and box with ropes was used to deploy them near the central line of the river. Once the entire setup reached its designated location, the readings on the Li-850 were recorded at 0.5 s intervals. During the entire measurement process, the box drifted freely with the current. The Li-850 was calibrated by the manufacturer before field trips. The rate of $CO_2$ efflux (FCO$_2$ in mmol m$^{-2}$ d$^{-1}$) was calculated from the observed change rate of the mole fraction S (ppm s$^{-1}$) using:

$$FCO_2 = (S \cdot V/A) \cdot t_1 \cdot t_2 \tag{5}$$

Where, S is the slope of $CO_2$ accumulation in the chamber (µatm s$^{-1}$), V is chamber gas volume (m$^3$), A is the chamber area (m$^2$), $t_1 = 8.64 \cdot 10^4$ s d$^{-1}$ is the conversion factor from seconds to days, and $t_2$ is a conversion factor from mole fraction (ppm) to concentration (mmol m$^{-3}$) at in situ temperature (T in K) and atmospheric pressure (p in Pa), according to the ideal gas law:

$$t_2 = p/(8.31 J K^{-1} mole^{-1} \cdot T) \cdot 1000 \tag{6}$$

The gas transfer velocity ($k$) was calculated from FCO$_2$ and $p$CO$_2$ in both water and ambient air using:

$$k = FCO_2/(K_0 \cdot (pCO_2^{water,i} - pCO_2^{air})) \tag{7}$$

To compare gas transfer velocity values among different sites, $k$ was standardized to $k_{600}$ as described by Alin et al. (2011) using:

$$k_{600} = k(600/Sc)^{-0.5} \tag{8}$$

Where, $Sc$ is the Schmidt number, which is dependent on temperature (T) in degree Celsius (Wanninkhof, 1992):

$$Sc = 1911.1 - 118.11T + 3.4527T^2 - 0.4132T^3 \tag{9}$$

In total, 196 chamber measurements were conducted. In 19 out of 215 sample sites, the drifting chamber was unable to deploy due to shallow water or high flow velocity. Meanwhile, 8 out of 196 $k_{600}$ data with the air−water $p$CO$_2$ gradient less than 200 µatm were also excluded, as the error in these calculations could be considerable (Borges et al., 2004).

## 3 Results

### 3.1 Physical and Biochemical Characteristics

The Dongjiang River was characterized by substantial seasonal variations in hydrologic regimes (Figure 2). Stream width in the wet season was 17.0 % and 5.6 % larger than that in the dry season for small and large rivers, respectively (Table S1). The discharge ranged 4 orders of magnitude from 0. 1 $m^3$ $s^{-1}$ in the small headwater streams during the dry season to 6690 $m^3$ $s^{-1}$ in the main stem during the wet season (Figure S1). Water temperature was higher in July and August (21.4–33 and 21–33.4 °C, respectively) than that in January (8.1–22.2 °C), April (16.5–26.9 °C), and October (17.4–29.7 °C). pH varied from 6.38 to 8.14, with a mean of 7.08. There was no significant (independent sample t test, $p > 0.05$) change in pH between wet and dry seasons. U10 based on all stream sites was higher in large rivers (0.86 ± 0.91 and 1.43 ± 1.58 m $s^{-1}$ in wet and dry season, respectively) than in small rivers (0.62 ± 0.61 and 0.76 ± 0.73 m $s^{-1}$ in wet and dry season, respectively).

The streams presented low alkalinity ranging from 225 to 3025 µmol $L^{-1}$. Overall, lower alkalinity was observed in wet season than in dry season (Table 1). In small rivers, the alkalinity in the wet season (656 ± 265 µmol $L^{-1}$) was 21.1 % lower than that in the dry season (831 ± 460 µmol $L^{-1}$), and the lowest alkalinity was observed in April (615 ± 262 µmol $L^{-1}$), which was 30.4 % lower than in January (883 ± 548 µmol $L^{-1}$). Similarly, the alkalinity in large rivers was 790 ± 402 µmol $L^{-1}$ in wet season, 14.5 % lower than 924 ± 411 µmol $L^{-1}$ in dry season. However, the lowest value of alkalinity in large rivers was observed in August (739 ± 312 µmol $L^{-1}$) instead of April in small rivers.

Spatial and seasonal changes in DOC concentration were also observed in the surveyed rivers (Table 1). DOC concentration in large rivers (1.94 ± 1.52 mg $L^{-1}$) was 41.6 % higher than that in small rivers (1.37 ± 0.72 mg $L^{-1}$). Meanwhile, DOC concentrations in the wet season were 2.22 ± 1.82 mg $L^{-1}$ and 1.54 ± 0.72 mg $L^{-1}$ for large and small rivers, respectively, which were 45.1 % and 54 % higher than that in the dry season (1.53 ± 0.72 and 1.11 ± 0.63 mg $L^{-1}$ for large and small rivers, respectively).

**Table 1** Seasonal Variations of Physical and Biochemical Characteristics, expressed as Mean ± SD.

| Stream size | Season | Month | Water Temperature (°C) | pH | Alkalinity ($\mu$mol L$^{-1}$) | DOC (mg L$^{-1}$) |
|---|---|---|---|---|---|---|
| *small* | Dry | January | 14.3 ± 4.1 | 7.05 ± 0.31 | 883 ± 548 | 1.07 ± 0.37 |
| | Wet | April | 19.9 ± 1.9 | 7.19 ± 0.26 | 615 ± 262 | 1.51 ± 0.58 |
| | Wet | July | 25.7 ± 2.3 | 7.17 ± 0.27 | 676 ± 227 | 1.59 ± 0.97 |
| | Wet | August | 27.1 ± 3.0 | 7.13 ± 0.38 | 678 ± 308 | 1.51 ± 0.56 |
| | Dry | October | 21.5 ± 2.6 | 7.08 ± 0.23 | 778 ± 358 | 1.16 ± 0.82 |
| *large* | Dry | January | 16.9 ± 5.5 | 7.00 ± 0.27 | 961 ± 409 | 1.70 ± 1.52 |
| | Wet | April | 22.1 ± 3.7 | 7.20 ± 0.27 | 890 ± 386 | 2.22 ± 1.65 |
| | Wet | July | 27.8 ± 2.9 | 6.92 ± 0.25 | 740 ± 305 | 1.97 ± 1.77 |
| | Wet | August | 28.9 ± 3.3 | 6.92 ± 0.26 | 739 ± 312 | 2.47 ± 2.04 |
| | Dry | October | 25.2 ± 3.1 | 7.13 ± 0.29 | 887 ± 331 | 1.37 ± 0.67 |

### 3.2 Spatial and Seasonal variations in $p$CO$_2$

The $p$CO$_2$ ranged from 15 to 6323 µatm with a catchment-wide average of 1748 µatm and showed considerable temporal and spatial variations throughout the sampling period. There was an increasing trend of observed $p$CO$_2$ from small to large rivers (Figure 3a). On average, the $p$CO$_2$ values were 856 ± 444, 1481 ± 979, 1354 ± 753, 2332 ± 1330, 2142 ± 1016, 2271 ± 1121, and 2168 ± 1046 µatm for streams from first to seventh order, respectively. The stronger increase in $p$CO$_2$ occurred between third and fourth order streams (from 1354 ± 753 to 2332 ± 1330 µatm, Figure 3a). Overall, $p$CO$_2$ in large rivers (2250 ± 1178 µatm) was 76.3 % higher than that in small rivers (1276 ± 796 µatm). Meanwhile, there was also an increasing trend of $p$CO$_2$ from rivers in the UDJRB compared with those in the LDJRB. The $p$CO$_2$ values were 2105 ± 959 and 2487 ± 1276 µatm for small and large rivers, respectively, in the LDJRB, which were 146.7% and 70% higher than that in the UDJRB, respectively (Figure 3b).

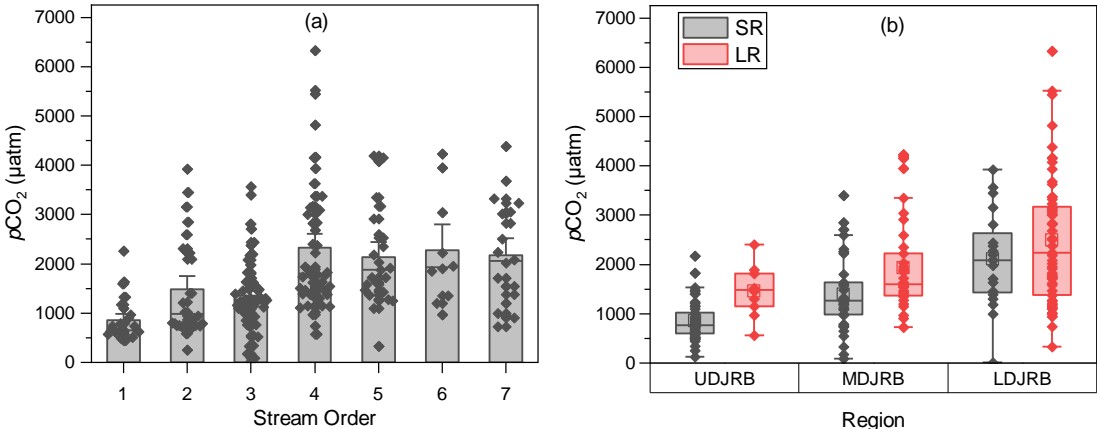

**Figure 3** Spatial variations in $pCO_2$. (a) Yearly average $pCO_2$ in the seven stream orders, standard errors (SE) are displayed by error bars. (b) Measured $pCO_2$ in small and large rivers among three regions in the DJRB. The box mid-lines represent medians; the interquartile range (IQR) is represented by top and bottom of the box, respectively; whiskers indicate the range of 1.5 IQR; the white square symbols represent means, and the other symbols represent $pCO_2$ values for each sampled site.

Seasonal variations of $pCO_2$ differed across the stream size spectrum (Figure 4). In small rivers, the highest $pCO_2$ was observed in April ($1506 \pm 880$ µatm), which was 50.3 % higher compared with January ($1002 \pm 660$ µatm). $pCO_2$ then decreased in July ($1131 \pm 589$ µatm) and increased in August ($1325 \pm 863$ µatm) and October ($1414 \pm 900$ µatm). Compared with small rivers, the peak of $pCO_2$ in large rivers occurred later but persisted for a longer period of time. In large rivers, an increase in $pCO_2$ was not observed until July. $pCO_2$ in April was $1831 \pm 793$ µatm, which was similar to $1805 \pm 1010$ µatm in January, and it increased 39.3 % to $2550 \pm 1210$ µatm in July. $pCO_2$ peaked in August ($2885 \pm 1351$ µatm) and then decreased to $2176 \pm 1166$ in October. Overall, $pCO_2$ was 9.3 % and 21.7 % higher in the wet season than in the dry season for small and large rivers, respectively.

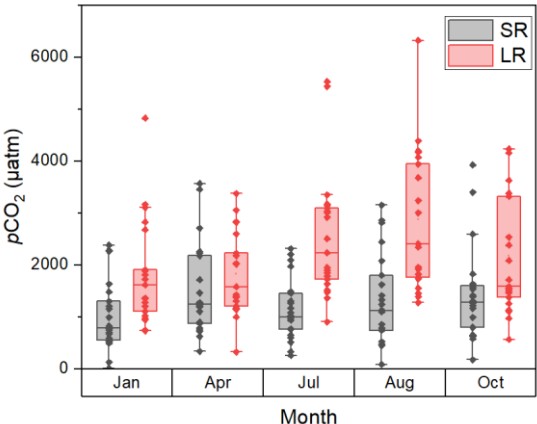

**Figure 4** Seasonal $pCO_2$ changes in small and large rivers. The box mid-lines represent medians; the interquartile range (IQR) is represented by top and bottom of the box, respectively; whiskers indicate the range of 1.5 IQR; the white square symbols represent means, and the other symbols represent $pCO_2$ values for each sampled site.

### 3.3 CO₂ effluxes and $k_{600}$

CO₂ effluxes ranged from $-129.8$ to $3874.8$ mmol m$^{-2}$ d$^{-1}$ with a mean of 225.2 mmol m$^{-2}$ d$^{-1}$. More than 95 % of the 196 samples had positive FCO₂ values, indicating that a majority of the surveyed rivers is a carbon source. Overall, we observed higher FCO₂ during wet season than during dry season in both small and large rivers (Figure 5a). FCO₂ in small rivers and large rivers were $264.2 \pm 410.0$ and $300.1 \pm 511.7$ mmol m$^{-2}$ d$^{-1}$, respectively, during the wet season, which was 87.2 % and 123.1 % higher than that in the dry season ($141.1 \pm 188.7$ and $134.5 \pm 129.5$ mmol m$^{-2}$ d$^{-1}$ for small and large rivers, respectively). No significant (independent sample t test, $p > 0.05$) difference in FCO₂ was observed between small and large rivers.

$k_{600}$ differed greatly between river size classes and among hydrological periods (Figure 5b). $k_{600}$ values in small rivers were on average significantly (independent sample t test, $p < 0.001$) higher than that in large rivers. The mean values of $k_{600}$ in small rivers were $8.29 \pm 11.29$ m d$^{-1}$ and $4.90 \pm 3.82$ m d$^{-1}$ for the wet season and dry season, respectively, which were 112.6 % and 70 % higher than that of large rivers ($3.90 \pm 5.55$ m d$^{-1}$ in the wet season and $2.25 \pm 1.61$ m d$^{-1}$ in the dry season). $k_{600}$ during the wet season were also significantly (independent sample t test, $p < 0.05$) higher than that in the dry season. $k_{600}$ increased 112.7 % and 118.2 % from dry season to wet season in small and large rivers, respectively.

However, comparisons between different phases in the same hydrological period (e.g., early, middle, and late wet season) did not differ significantly (paired sample t test, $p > 0.05$) for both river size classes.

The spatial and temporal variations of $CO_2$ efflux generally coincided with the changes in $pCO_2$ and $k_{600}$. In small rivers, the highest $CO_2$ effluxes were $346.8 \pm 625.2$ mmol m$^{-2}$ d$^{-1}$ during April, consistent with the high $k_{600}$ and $pCO_2$ in this period. In large rivers, high $CO_2$ effluxes were observed in both April ($339.9 \pm 828.6$ mmol m$^{-2}$ d$^{-1}$) and August ($329.9 \pm 270.0$ mmol m$^{-2}$ d$^{-1}$), which were attributed to the concurrently high $k_{600}$ in April and high $pCO_2$.

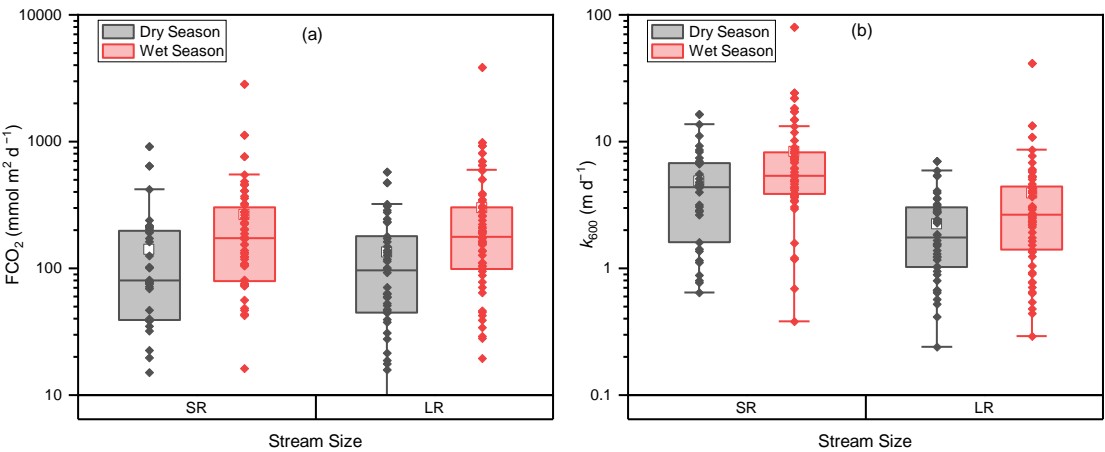

**Figure 5** Relationship between stream size and (a) FCO$_2$ and (b) $k_{600}$. The box mid-lines represent medians; the interquartile range (IQR) is represented by top and bottom of the box, respectively; whiskers indicate the range of 1.5 IQR; the white square symbols represent means, and the other symbols represent FCO$_2$ and $k_{600}$ values for each sampled site.

## 4 Discussions

### 4.1 Underlying Processes of $p$CO$_2$ dynamics

The spatial pattern of $pCO_2$ in the DJRB is likely resulting from the changes in terrestrial carbon inputs (i.e., organic and inorganic carbon) and in-stream metabolism, both of which varied due to different land cover and catchment topography. The higher $pCO_2$ values in large rivers than small rivers were associated with a higher percentage of urban and cropland cover and a lower forest cover (Figure 6). Compared with forest, cropland could provide a more favourable condition for soil erosion and the transfer of terrestrial carbon from land to rivers, contributing to a higher $pCO_2$. Intensification of

agricultural practices could promote the decomposition of soil organic matter (Borges et al., 2018), thereby increasing the concentration of $CO_2$ and liable DOC in the soil (Borges et al., 2018). The soil $CO_2$ could be easily transported to rivers and thus increase the $pCO_2$, while the liable DOC could be decomposed rapidly after entering the rivers due to their sensitivity to in-stream metabolism (Lambert et al., 2017; Li et al., 2019). Meanwhile, the input of wastewater with high organic matter concentration from urban areas could also contribute to an increase in riverine $pCO_2$ (Xuan et al., 2020; Zhang et al., 2021). Our results showed increasing $pCO_2$ from forest-dominated streams in the UDJRB relative to those in agricultural and urban impacted catchments in the MDJRB and LDJRB (Figure 3b). The >70% forest cover in the UDJRB (Figure 1) may have greatly reduced the soil erosion intensity (Ran et al., 2018). Meanwhile, the organic matter from forest tends to be more aromatic, thus more capable of surviving biodegradation (Kalbitz and Kaiser, 2008), leading to a relatively low riverine $pCO_2$ value. In contrast, cropland, occupying about 49% of the land cover (Figure 1), was the primary land use type in the MDJRB substituting forest, and urban areas account for ~17% of the land cover in the LDJRB. The higher $pCO_2$ in the MDJRB and LDJRB is likely under the influence of agricultural practices and wastewater input. Overall, land use mainly affects the spatial distribution of $pCO_2$ by altering the amount and lability of carbon inputs to the rivers.

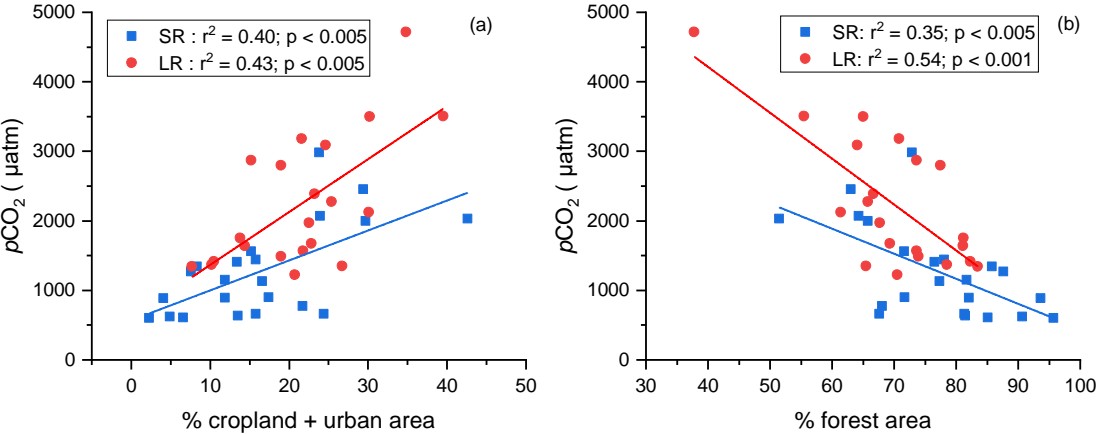

Figure 6 (a) the relationship between yearly average $pCO_2$ at each site and the percentage of cropland and urban area combined, (b) the relationship between yearly average $pCO_2$ at each site and the percentage of forest area

Moreover, different catchment topography in small and large rivers may have also contributed to the differences in $p\mathrm{CO_2}$. Due to steeper channel slopes and higher flow velocities, small rivers in the DJRB have higher k600 (Figure 5b). As a consequence, $\mathrm{CO_2}$ in small rivers can exchange with the atmosphere more rapidly, preventing the build-up of dissolved $\mathrm{CO_2}$ and thus lower $p\mathrm{CO_2}$ (Rocher-Ros et al., 2019). Therefore, other processes have facilitated the carbon transfer from small rivers to downstream large rivers, sustaining the higher $p\mathrm{CO_2}$ in large rivers. Recent studies indicate that carbonate buffering could decrease the $\mathrm{CO_2}$ emissions from small rivers by increasing the ionization of $\mathrm{CO_2}$ (Stets et al., 2017), thereby increasing the transfer of DIC towards the rivers downstream, which resulted in the higher $p\mathrm{CO_2}$ in downstream large rivers . However, strong carbonate buffering usually occurs in high-alkalinity ($>2500$ μmol $L^{-1}$) streams with high pH ($>8$), while in low-alkalinity waters, the pool of ionized $\mathrm{CO_2}$ is relatively small, indicating a weak carbonate buffering (Stets et al., 2017). Since the streams in the DJRB were characterized by low alkalinity ($726 \pm 364$ μmol $L^{-1}$ and $844 \pm 409$ μmol $L^{-1}$ for small and large rivers, respectively), carbonate buffering is unlikely a primary contributor to the high $p\mathrm{CO_2}$ in large rivers. Meanwhile, our data showed that river water $p\mathrm{CO_2}$ was negatively related to DO and positively related to DOC (Figure 7), suggesting that the high $p\mathrm{CO_2}$ in large river was related to metabolic processes. The steep channel slopes in small rivers tend to promote the transfer of OC to downstream large rivers. As a consequence, it is difficult for terrestrial organic carbon to be converted into $\mathrm{CO_2}$ in small rivers due to the short water residence time (Hotchkiss et al., 2015). Conversely, a greater fraction of OC may have been transported downstream and fuel the heterotrophic respiration in large rivers, where low flow velocity and long water residence time facilitated the decomposition of organic carbon within the water column (Denfeld et al., 2013).

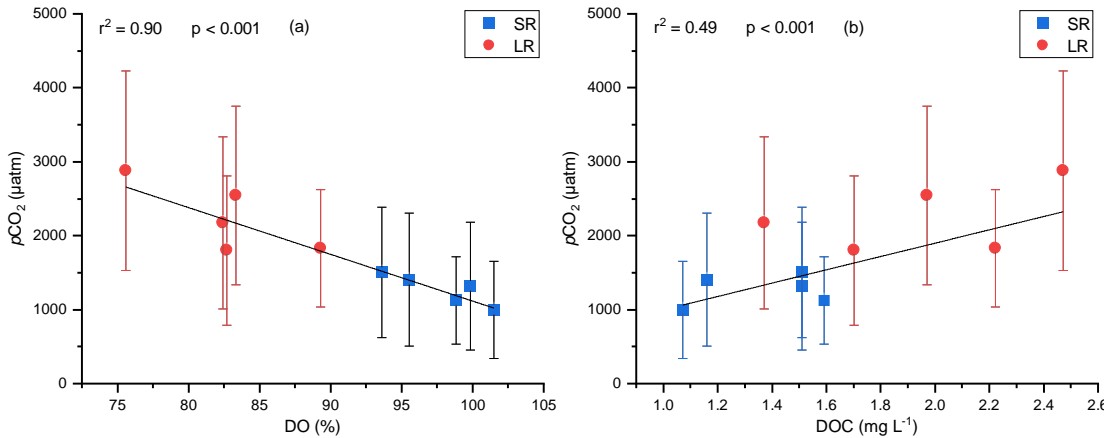

**Figure 7** Relationship between seasonal average $pCO_2$ and (a) DO and (b) DOC. Error bars for the $pCO_2$ represent 1 standard deviation from the seasonal mean. The DO–$pCO_2$ and DOC–$pCO_2$ relationship are shown as solid lines.

To compare the contribution of internal metabolism and external $CO_2$ input on $pCO_2$ in small and large rivers, the $\Delta CO_2{:}\Delta O_2$ stoichiometry was used to evaluate the impacts of respiration and photosynthesis processes on the concentration of dissolved $O_2$ and $CO_2$ (Stets et al., 2017). The inverse relation between $\Delta CO_2$ and $\Delta O_2$ (Figure 8) demonstrated that metabolic processes are important for the dissolved $CO_2$ concentration variations (Amaral et al., 2020), while the difference in the $\Delta CO_2{:}\Delta O_2$ stoichiometry between small and large rivers suggested the different strength of in-stream metabolism (Rasera et al., 2013). The $\Delta CO_2{:}\Delta O_2$ stoichiometry in large rivers is closer to the 1:1 line than that in small rivers, indicating that large rivers are more affected by the metabolic processes (Jeffrey et al., 2018; Amaral et al., 2020). For large rivers, the linear regression is $\Delta CO_2 = -0.999\ (\pm 0.081)\ \Delta O_2 + 18.020\ (\pm 5.995)$ ($r^2 = 0.62$, $p < 0.001$). When the $CO_2$ concentration increases in large rivers, a similar magnitude of decrease in dissolved $O_2$ concentration occurs, indicating that in-stream metabolism is the primary control on $pCO_2$. In contrast, the linear regression for small rivers is $\Delta CO_2 = -0.868\ (\pm 0.098)\ \Delta O_2 + 21.42\ (\pm 4.175)$ ($r^2 = 0.41$, $p < 0.001$), which means that with the $CO_2$ concentration increasing by 1 µmol $L^{-1}$, the $O_2$ concentration decreases by only 0.868 µmol $L^{-1}$. Therefore, extra $CO_2$ inputs have contributed to the changes in $pCO_2$ despite the strong presence of in-stream metabolism.

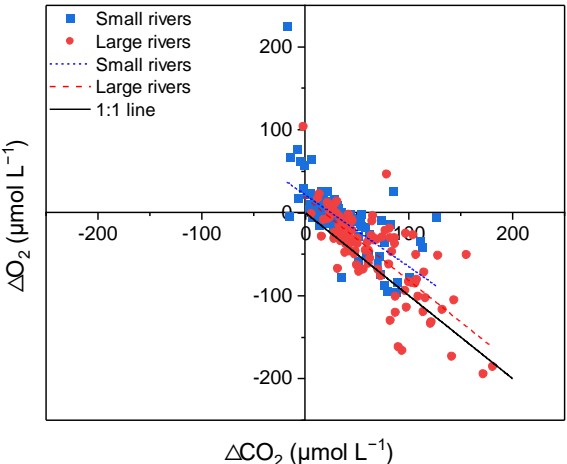

**Figure 8** The relationship between $\Delta CO_2$ and $\Delta O_2$. Points greater than zero are oversaturated, and less than zero are undersaturated. Points above the 1:1 line indicate the existence of additional carbon sources, apart from in-stream metabolic processes. For large rivers, the linear regression is $\Delta CO_2 = -0.999\ (\pm 0.081)\ \Delta O_2 +18.020\ (\pm 5.995)$ ($r^2 = 0.62$, $p < 0.001$). For small rivers, the linear regression is $\Delta CO_2 = -0.868\ (\pm 0.098)\ \Delta O_2 + 21.42\ (\pm 4.175)$ ($r^2 = 0.41$, $p < 0.001$).

On the other hand, the temporal pattern was affected by precipitation and temperature seasonality. Our results showed that higher $pCO_2$ occurred in the wet season than in the dry season for both small and large rivers (Figure 4). The elevated temperature in the wet season could promote a substantial increase in the net primary productivity of the terrestrial ecosystem, while increased precipitation can facilitate the transfer of terrestrial carbon (Rasera et al., 2013), including both soil $CO_2$ and OC, from land to rivers. This could either directly increase riverine $pCO_2$, or fuel OC decomposition (Borges et al., 2018). However, the differences in seasonal changes of $pCO_2$ between small and large rivers (Figure 4) also suggested that their controlling process could be different. For small rivers, the highest $pCO_2$ value was observed in April (Figure 4), which is consistent with the rapid surge of terrestrial C inputs, usually occurring at the onset of the wet season (Hope et al., 2004; Yao et al., 2007; Johnson et al., 2008). However, such increase in $pCO_2$ was not observed in large rivers (Figure 4) though the DOC in large rivers increased at a rate similar to that in small rivers during the same period (Table 1). A possible explanation is that the observed $pCO_2$ rise was mainly originated from soil $CO_2$, which was readily emitted from the small rivers into the air, with little reaching the larger rivers downstream (Denfeld et al., 2013; Drake et al., 2018). Differences in the $pCO_2$ dynamics in July and August also reflected different controlling processes in small and large rivers. A decline in $pCO_2$ in July in small rivers

suggested that it might have experienced the depletion effect occurring in the middle and late wet season
(Hope et al., 2004), during which soil $CO_2$ decreased due to the continual precipitation. In contrast, the increase in $pCO_2$ in large rivers in July indicated that the decreased soil $CO_2$ inputs could hardly affect the $pCO_2$ in large rivers during this period. Instead, stronger in-stream metabolism caused by OC inputs and the favourable conditions for OC decomposition are more likely to be responsible for the rising $pCO_2$. In addition, there are other processes that may have affected the riverine $pCO_2$. For example, stronger
solar radiation during summer could increase photo-oxidation in rivers. However, the commonly observed lower daytime $CO_2$ emission rates than nocturnal rates (Gómez-Gener et al., 2021) suggest that photosynthesis overrides photo-oxidation in $CO_2$ dynamics. Nonetheless, the low DO concentration observed in the surveyed rivers (Figure 8) suggested that photosynthesis is not likely the primary control on the seasonal variation of $pCO_2$.

**4.2 Environmental Control of $k_{600}$ variation**

Environmental factors, including wind speed and hydrological variables, could affect the gas exchange at the water–air interface and are typically used to explain the variance in $k_{600}$ (Alin et al., 2011; Raymond et al., 2012). Flow velocity generally determines the $k_{600}$ in small rivers, while wind speed becomes a more important factor in controlling the $k_{600}$ in large rivers, reservoirs, and estuary (Guérin et al., 2007;
Rasera et al., 2013; Amaral et al., 2020). In our surveyed rivers, $k_{600}$ displayed a significant linear correlation (Pearson correlation, $p < 0.001$) with the flow velocity. Our $k_{600}$ model (Figure 9) based on 188 field measurement data is similar to that developed by Alin et al. (2011) ($k_{600} = 13.82 + 0.35v$). However, in our studied rivers, no significant correlation (Pearson correlation,$p > 0.05$) was found between wind speed and $k_{600}$ regardless of stream size. This could be explained by the lower wind speed
($0.68 \pm 0.66$ m s$^{-1}$ and $1.09 \pm 1.06$ m s$^{-1}$ for small and large rivers, respectively; Table 2) (Guérin et al., 2007). As the wind speed decreases, the impact of flow velocity on $k_{600}$ becomes increasingly predominant (Borges et al., 2004). Therefore, the accuracy of $k_{600}$ estimation based on wind speed in nearby regions should be examined using measurement data (Yao et al., 2007; Li et al., 2018). The temporal heterogeneities of $k_{600}$ between small and large rivers reveal the differences in flow regime. The
$k_{600}$ in small rivers are significantly higher than that in large rivers (independent sample t test, $p < 0.001$), which could be explained by the higher flow velocity in small rivers. Meanwhile, the significantly higher $k_{600}$ in the wet season than in the dry season (independent sample t test, $p < 0.05$) is the result of the

increased flow velocity and turbulence due to monsoon-induced precipitation during the wet season (Guérin et al., 2007; Alin et al., 2011; Ho et al., 2018).

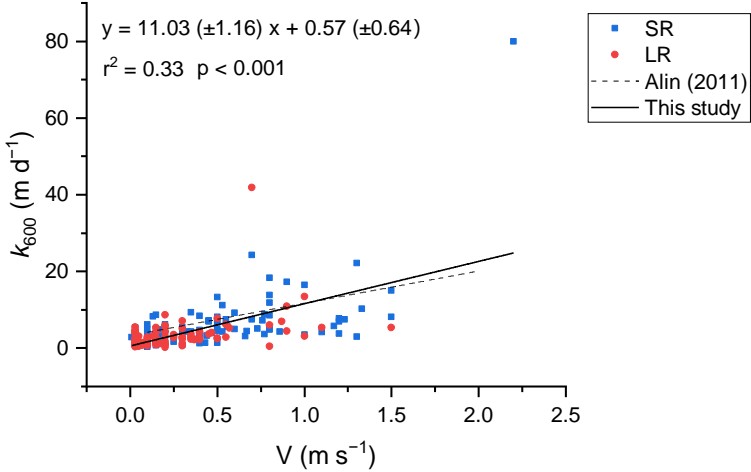

**Figure 9** Relationship between $k_{600}$ and flow velocity. The dashed line represents the parameterization of Alin et al (2011).

**Table 2.** Seasonal variation of $k_{600}$ and environmental factors in small and large rivers.

| Stream size | Season | Current velocity (m s$^{-1}$) | U10 (m s$^{-1}$) | $k_{600}$ (m d$^{-1}$) |
|---|---|---|---|---|
| small | Wet | 0.66 ± 0.47 | 0.62 ± 0.61 | 8.29 ± 11.29 |
| | Dry | 0.43 ± 0.27 | 0.76 ± 0.73 | 4.90 ± 3.82 |
| large | Wet | 0.32 ± 0.32 | 0.86 ± 0.91 | 3.90 ± 5.55 |
| | Dry | 0.17 ± 0.19 | 1.43 ± 1.58 | 2.25 ± 1.61 |

Exceptionally high $k_{600}$ values were observed in the surveyed rivers (Figure 9). The highest $k_{600}$ in large and small rivers were 41.83 and 79.97 m d$^{-1}$, respectively, which were 5-fold and 3-fold larger than calculated $k_{600}$, respectively. This is likely the result of the exponential increase in $k_{600}$ due to extreme flood events. Generally, flood events associated with heavy rainfall can substantially increase flow velocity and near-surface turbulence (Almeida et al., 2017; Geeraert et al., 2017), leading to extremely high $k_{600}$ values. Yet, neither our model nor the one from Alin et al. (2011) was suitable for the estimation of $k_{600}$ during extreme flood events because the calculated $k_{600}$ could deviate far from the measured $k_{600}$ when they occurred. The extent to which flood events affect $k_{600}$ and riverine $CO_2$ emission is still uncertain and warrant continued research (Drake et al., 2018).

### 4.3 A Comparison of $CO_2$ Emissions to Other Rivers

The mean $CO_2$ fluxes of 225.2 mmol $m^{-2}\,d^{-1}$ in the DJRB is comparable to those observed in tropical
and subtropical rivers in the Americas, Africa, and Southeast Asia (Table 3). Although the magnitude
of the $CO_2$ emissions of these river systems is similar, the seasonal variations and drivers behind them
could differ. The $CO_2$ emissions from the Dongjiang River were higher in the wet season than in the dry
season. This seasonal pattern is similar to that observed in the Xijiang and Daning rivers (Yao et al.,
2007; Ni et al., 2019) but different from that observed in the Jinshui River in the upper Yangtze River,
where $p$CO$_2$ is high in winter and low in summer (Luo et al., 2019), although all four rivers are in the
East Asia Monsoon climate region. The seasonal differences in $CO_2$ emissions are largely caused by the
$p$CO$_2$ variability, which in turn is regulated by external carbon inputs, internal production of $CO_2$ (Yao
et al., 2007), and the dilution effect caused by precipitation (Johnson et al., 2007). For rivers where $p$CO$_2$
is lower in summer than in winter, the dilution effect overrides the effect of increased carbon inputs and
internal $CO_2$ production (Luo et al., 2019). In contrast, for rivers like the Dongjiang River, although the
dilution effect remains, increased $CO_2$ inputs and metabolism are more significant factors in controlling
its $p$CO$_2$, thus leading to higher summer $p$CO$_2$. In addition, the controlling processes of the Dongjiang
River could be different even when compared with rivers with similar seasonal variations in the same
climatic zone. For instance, the DO in the Xijiang river was supersaturated, indicating that its aquatic
photosynthetic activities predominated aquatic metabolism and tended to reduce its $CO_2$ concentration
(Yao et al., 2007). Therefore, other carbon sources like soil respiration and carbonate weathering should
be responsible for the high $p$CO$_2$ in summer (Zhang et al., 2019). In contrast, the low DO value and the
negative correlation between DO and $p$CO$_2$ in the Dongjiang River indicated that photosynthesis is
relatively weak compared with the respiration, and the latter process is an essential source of riverine
$CO_2$ (Stets et al., 2017), resulting in a higher $p$CO$_2$ in summer.

**Table 3.** Comparison of $CO_2$ emissions from subtropical and tropical rivers.

| Rivers | Climate | Season | $pCO_2$ (µatm) | $k_{600}$ (m d$^{-1}$) | $FCO_2$ (mmol m$^{-2}$ d$^{-1}$) | References |
|---|---|---|---|---|---|---|
| The Dongjiang River (Large rivers) | Subtropical | Wet | 2422 ± 1209 | 3.90 ± 5.55 | 300.1 ± 511.8 | This study |
| | | Dry | 1990 ± 1094 | 2.25 ± 1.61 | 134.5 ± 129.5 | |
| The Dongjiang River (small rivers) | | Wet | 1321 ± 792 | 8.29 ± 11.29 | 264.2 ± 410.0 | |
| | | Dry | 1191 ± 825 | 4.90 ± 3.82 | 129.5 ± 197.2 | |
| The Xijiang River (Mainstream) | Subtropical | | 2600 | | 190.3–358.6 | (Yao et al., 2007) |
| The Lower Mekong River | Tropical | | 1090 ± 290 | 6.24* | 194.5 | (Li et al., 2013) |
| The Yangtze River ( Jinshui River) (headwater stream) | Subtropical | | 1147 ± 874 | 11.1 ± 4.5* | 343 ± 413 | (Luo et al., 2019) |
| | | Dry | 1562 ± 975 | | 542 ± 477 | |
| | | Wet | 834 ± 639 | | 192 ± 278 | |
| The upper Yangtze River (Daning river) | Subtropical | | 1198.2 ± 1122.9 | | 329.8 ± 470.2 | (Ni et al., 2019) |
| | | Rainy | 1243.7 ± 1111.5 | 8.1–14.1* | 357.4 ± 483.7 | |
| | | Dry | 1145.5 ± 1146.2 | 7.0–8.8* | 288.7 ± 450.0 | |
| The Zambezi River | Tropical | Wet | 3102.5 ** | 0.05–1.51 | 350.75 | (Teodoru et al., 2014) |
| | | Dry | 1150 ** | | 51.92 | |
| The Congo River | Tropical | High water | 6001 ± 5008 | | 1149 or 1520 | (Borges et al., 2015a; Borges et al., 2015b) |
| | | Low water | 4867 ± 2578 | | | |
| | | Falling water | 5321 ± 3383 | | | |
| The Lower Red River | Tropical | | 1589 ± 43 | 12.22 ± 6.48 | 530.3 ± 16.9 | (Le et al., 2018) |
| Caboolture River | Subtropical | | 3000 ± 33 | | 379 ± 53 | (Jeffrey et al., 2018) |
| Rajang River | Tropical | wet | 2531 ± 188 | 0.55–2.93 | 141.67 | (Müller-Dum et al., 2019) |
| | | dry | 2337 ± 304 | | 125 | |
| Lower Mississippi River | Subtropical | | 1514 ± 652 | | 172.8 | (Reiman and Xu, 2019b) |
| Amazonian Rivers | Tropical | | 259–7808 | 5.06 | 69.12–1321.92 | (Rasera et al., 2013) |

* $k$ values wereshown here because $k_{600}$ values were not provided in references; ** the unit for $pCO_2$ is ppm.

The $CO_2$ fluxes in small rivers are similar to those in large rivers, which is contradictory to the finding in previous studies that $CO_2$ effluxes should be higher in small rivers than in large rivers due to the input of $CO_2$-rich groundwater (Duvert et al., 2018). The depletion and diffusion effect may be responsible for the discrepancy (Johnson et al., 2007; Dinsmore et al., 2013). Groundwater in the DJRB could be easily diluted due to abundant monsoon-induced rainfall, preventing it from supplying the small rivers with

high $CO_2$ concentrations. However, we recognize that the impact of groundwater on $pCO_2$ in small rivers may be overlooked in our sampling process since the $CO_2$ carried by groundwater can emit into the atmosphere within a very short distance (Duvert et al., 2018). In view of the above, it is recommended that further studies targeting the release of groundwater $CO_2$ to the atmosphere be carried out in the future.

**5 Conclusion**

Studying $CO_2$ emissions from subtropical rivers is an essential step toward more accurate estimates of global $CO_2$ emissions from river systems. By deploying floating chambers, seasonal changes in riverine $pCO_2$ and $CO_2$ emissions from the Dongjiang River catchment were investigated. Spatial and temporal patterns of $pCO_2$ were mainly affected by terrestrial carbon inputs (i.e., organic and inorganic carbon) and in-stream metabolism, both of which varied due to different land cover, catchment topography, and

seasonality of precipitation and temperature. $k_{600}$ was higher in small rivers than in large rivers and higher during the wet season than during the dry season, both of which can be explained by the observed significant correlation between $k_{600}$ and flow velocity. In contrast to previous studies, similar $CO_2$ fluxes were observed among small and large rivers in the DJRB. It is suggested that the absence of commonly observed higher $CO_2$ fluxes in small rivers could be associated with the depletion effect caused by

abundant and persistent precipitation in this subtropical monsoon catchment. There is no doubt that the spatial and temporal variations of $CO_2$ emissions from the DJRB reflected the complexity and diversity of controlling factors. As a step towards a more accurate estimate of the carbon budget in the catchment, comprehensive and systematic measurements of $CO_2$ emissions covering a broad range of stream sizes and seasons are of paramount importance.

*Data availability.* $CO_2$ emission data used in this study are available online at: https://doi.org/ 10.25442/hku.13416281.v1 (Liu, 2020). Other data are available from the corresponding author Lishan Ran upon request at lsran@hku.hk.

*Author contributions.* BL and LR conceived the study. BL, MT, CC, XY, and LR carried out the fieldwork. BL, MT, and KS designed and performed the laboratory analysis. BL composed the
manuscript with contributions from all authors.

*Competing interests.* The authors declare that they have no conflict of interest.

*Acknowledgements.* This work was financially supported by the Research Grants Council of Hong Kong (Grants: 17300619 and 27300118), the Hui Oi-Chow Trust Fund (Grant: 201801172006) and the National Natural Science Foundation of China (Grant: 41807318). We thank Dr Steven Bouillon and the
two anonymous reviewers for their constructive comments which have greatly improved the manuscript.

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
