# Peer review of "Spatial and temporal variability of $p\text{CO}_2$ and $\text{CO}_2$ emissions from the Dongjiang River in South China"

_Biogeosciences, 2020_

## Author Response (AR1)

**Referee #1**

The paper by Liu et al. presents seasonal $pCO_2$ concentrations and $CO_2$ fluxes from the Dongjiang River basin. They found that concentrations and fluxes were higher in larger rivers relative to smaller ones and in the wet season (summer) compared to the dry season. They also contextualized some of these broader findings with auxiliary measurements of DO, DOC, alkalinity, and pH.

The paper is presents a good quantity of spatially and temporally resolved $CO_2$ data, adheres to established methods, and is generally well written. I think the data alone is a useful contribution however I think much of the discussion surrounding the drivers and explanation of $CO_2$ differences is either lacking or unsupported. I think that after some revisions of the discussion, the manuscript could warrant publication in Biogeosciences. Below are my primary criticisms, followed by line-specific minor comments.

**Primary criticisms**

The results show that $pCO_2$ (and in turn $FCO_2$) is higher in the larger rivers compared to the smaller rivers, which the authors interpret as resulting from proportional differences in C inputs (both $CO_2$ and DOC) and metabolism of allochthonous inputs. Since these are connected systems (i.e. the small rivers eventually flow into the larger ones). I'm a bit puzzled how $CO_2$ would increase downstream due to higher C inputs unless the study design somehow missed high $CO_2$ inputs from low order streams that directly joined the mainstream?

Based on Figure 1, it appears that many of the smaller rivers were also at higher elevation. A bias towards higher altitude sites in the smaller rivers could explain the observed trends if these catchments had less vegetation/forest cover and therefore less C inputs (as both $CO_2$ and DOC). Indeed, the authors observed higher DOC concentrations in larger rivers, which they assume fuels higher respiration. Where does this DOC come from if it doesn't pass through smaller rivers first? I suspect there is some sampling bias at hand.

Reply: We have observed higher $pCO_2$ in large rivers compared with the smaller rivers (Figure 3a and Figure 4 in the revised manucript) and interpreted it as resulting from stronger in-stream metabolism of allochthonous C inputs. However, stronger in-stream metabolism may caused by increased C inputs or more favorable conditions for OC decomposition. The latter is more likely to be responsible for the spatial pattern of $pCO_2$ in this study. Small rivers located in hill-dominated Dongjiang River Basin (DJRB) tend to have high flow velocity and short water residence time, making it hard for terrestrial organic carbons to convert into $CO_2$ (Hotchkiss et al., 2015). Meanwhile, long water residence time in large rivers could facilitate the decomposition of organic carbon within the water column (Denfeld et al., 2013). Several missed high $CO_2$ inputs from low order streams that directly join the mainstream is unlikely to be responsible for the high $pCO_2$ due to their relatively small discharge.

We realized the higher DOC in large rivers than that in upstream small rivers and suggested that there may have extra DOC inputs other than direct inputs from small rivers. Therefore, based on the referee's suggestion, we examined the impact of land use on riverine $pCO_2$. Our result showed increased $pCO_2$ from forest-dominated streams in upper DRJB to Agricultural and urban impacted catchments in middle DJRB and lower DJRB (Figure 1 and Figure 3b). Agricultural

practices on cropland and domestic wastewater from the urban areas could contribute to the higher DOC and thus higher $p\text{CO}_2$ as observed in large downstream rivers compared with small rivers. However, the effect of increased DOC on $p\text{CO}_2$ is confined by other factors that control the OC decomposition. Our results showed that large rivers had similar DOC concentration but higher $p\text{CO}_2$ compared with small rivers with similar land cover (Figure 7 in the revised version of manucript), when the percentage of forest area was over 65% or when the percentage of the combined area of cropland and urban area was less than 30%, which was the case for the majority of our sample sites. This indicated that in-stream metabolism was stronger in large rivers than in small rivers despite similar DOC concentrations. Less C inputs due to more forest cover is not enough to explain the lower $p\text{CO}_2$ in small rivers. Please refer Line 240–244 and L305-330 in the revised version of the manuscript for the changes.

There are additionally more processes, such as photo-oxidation or titration of the carbonate equilibrium via organic acids (indeed you see increasing $\text{CO}_2$ with decreasing alkalinity), that could impact some the observed downstream increase in $\text{CO}_2$. These aspects are not discussed in the manuscript and the authors conclude too strongly that they know the responsible drivers without data to support such claims. Since more highly productive vegetation in the catchment could result in both higher $\text{CO}_2$ inputs and higher DOC that fuels respiration, I think it would be useful to explore the relationship between C concentrations ($p\text{CO}_2$ and DOC) and catchment land-cover (perhaps as a fraction of wet area, similar to Rocher-Ros et al 2019, L&O Letters or % forest cover).

Reply: Thank you for your suggestion. We have examined the impact of land use on DOC input and riverine $p\text{CO}_2$. Our result showed that both DOC and $p\text{CO}_2$ were negatively related to the percentage of forest area and positively related to the percentage of cropland and urban area combined (Figure 8 in the revised version of the manuscript). Agriculture practiceson cropland and domestic wastewater from the urban area could contribute to the higher DOC and $p\text{CO}_2$ as observed in large downstream rivers compared to upstream small rivers. We also observed increased $p\text{CO}_2$ from forest-dominated streams in UDJRB to the agricultural and urban impacted catchment in MDJRB and LDJRB (Figure 1 and Figure 3b in the revised manuscript). Moreover, our results showed that large rivers had similar DOC concentration but higher $p\text{CO}_2$ compared with small rivers with similar land cover (Figure 7 in the revised manuscript ), when the percentage of forest area was over 65% or when the percentage of the combined area of cropland and urban area was less than 30%, which was the case for the majority of our sample sites. This suggests that in-stream metabolism was stronger in large rivers than in small rivers despite similar DOC concentrations, which also supported our arguments about the favorable condition for OC decomposition in large rivers. Please refer to Lines 307–359 in the revised version of the manuscript for the changes.

We have added two Figures (Figure 1 and Figure 7, Line 102 and Line 335 in the revised version of the manuscript) to support the discussion about the relationship between land cover and C concentration.

[Figure]

90 **Figure 1** Sample sites and land cover in the DJRB. Yearly average $p$CO$_2$ at each sample site was displayed. Based on the land cover dataset: FROM-GLC10 (http://data.ess.tsinghua.edu.cn). Note, this figure has also been added into the revised version of the manuscript.

[Figure]

95

**Figure 7** (a) the relationship between yearly average $pCO_2$ at each site and the percentage of cropland and urban area combined (b) the relationship between yearly average $pCO_2$ at each site and the percentage of forest area (c) the relationship between yearly average DOC at each site and the percentage of cropland and urban area combined (d) the relationship between yearly average DOC at each site and the percentage of forest area. Note, this figure has also been added into the revised version of the manuscript.

100

The discussion of spatial and temporal patterns is blended together and needs to be disambiguated a bit. It is hard for the reader to make sense of these various overlapping trends. I would suggest starting with one (spatial), then the other (temporal) before finishing on how they overlap to result in the observed pattern.

105

Reply: Thank you for your advice. We have revised the discussion about the drivers of spatial and temporal pattern. We started with analyzing the possible major controls of spatial pattern , then the temporal pattern, before finishing on other minor controlling factors. Please refer to Lines 293–380 in the revised version of the manuscript for the changes.

110

Increased precipitation can both increase the transport of terrestrial C (including $CO_2$) and dilute it. How do you know which process dominates?

115

Reply: We estimated the impact of those effects by analyzing the temporal pattern of precipitation and riverine $CO_2$. For example, precipitation and $CO_2$ concentration increased simultaneously from January to April in small rivers, which suggested that the increased terrestrial C transport was likely the dominant process in controlling the $pCO_2$ changes. In contrast, the precipitation was similar in April and July, but the $CO_2$ concentration decreased from April to July. Thus, the dilution and depletion effects caused by precipitation was more obvious. We also elaborated it in Lines 344–355 in the revised version of the manuscript.

Throughout the discussion, the authors fail to reference their figures or tables in many cases that would make it much easier to observe their explanations.

Reply: Thank you for your advice. We have improved the referencing of figures to support our arguments.

Given the high resolution of the $pCO_2$ data, would it not be interesting to upscale outgassing for the whole basin? Perhaps it could be compared to DOC/POC export if those have been previously estimated (or even roughly estimated using your values). At the very least, I think the authors' data could be nicely displayed on a map (Similar to Figure 1 of Rocher-Ros 2019, Limnology and Oceanography Letters)

Reply: We are also very interested in the calculation of basin-wide $CO_2$ emissions. One of our objectives was to provide support for more accurate global $CO_2$ emission estimates. Currently, the estimation of $CO_2$ emissions at the watershed scale is limited by the accuracy of the $CO_2$ data and the surface area accuracy of the river networks. We are now working on a parallel study about the estimates of basin-wide $CO_2$ emissions from the river network in the Dongjiang river basin. We intend to perform higher-precision river network extraction and water area calculations by combining remote sensing images and DEM (Wang et al., 2018; Wang et al., 2020). A more accurate watershed-scale carbon emissions estimation could then be carried out. Moreover, we will also compare it with lateral carbon export and NPP. In this study, we intend to focus on the factors that regulate the difference in $CO_2$ concentration and emissions between large and small rivers. Thank you for your advice about the data presentation. In addition, $pCO_2$ and land use cover have been displayed in Figure 1 (please refer to Line 101 in the revised version of manuscript for the changes)

Overall, I think the discussion of the drivers of $CO_2$ variability is overstated. Specifically, there is no direct evidence of lateral soil $CO_2$ nor dilution effect caused by precipitation.
There doesn't seem to be much of a difference in $dCO_2$ vs. $dO_2$ between large and small rivers (Figure 6), suggesting that metabolism is similar.
At minimum, the current discussion would need to justify why simultaneously low DOC and $CO_2$ are not an artifact of altitude/land-cover.

Reply: Thank you very much for the comments and suggestions. We fully agree with the referee's comments. Based on your comments, we have substantially revised the manuscript by re-discussing the potential impact of other factors, including land use cover (Lines 305–334), , photo-oxidation and photosynthesis(Lines 375–380 in the revised version of the manuscript ).

Regarding the result of $dCO_2$ vs. $dO_2$, a negative relation between $dCO_2$ vs. $dO_2$ was observed in both small and large rivers, suggesting that metabolism have occurred in both of them. However, $dO_2$ is higher in small rivers compared with large rivers with similar $dCO_2$, suggesting that the impacts of other factors other than metabolism should be more important in small rivers, which is also supported by the higher $pCO_2$ in large rivers than in small rivers with similar land cover and DOC concentration (Figure 8 in the revised version of the manuscript).

Based on the referee's suggestion, we have also examined the impact of land use on DOC input and riverine $pCO_2$ (Lines 324–334 in the the revised version of the manuscript). Our results showed that large rivers had similar DOC concentration but higher $pCO_2$ compared with small rivers with similar land cover (Figure 7 in the the revised version of the manuscript). This indicates that the higher $pCO_2$ in large rivers is more likely the result of favorable conditions for OC decomposition, even though higher DOC could contribute to higher $pCO_2$. Therefore, the observed spatial pattern is not the result of sampling bias. Moreover, both small and large rivers with a variety of land cover structures were sampled in this study to reduce the possible bias caused by the distribution of sample sites (Figure S4, also available in Supplementary).

[Figure]

**Figure S4** Forest, cropland, and urban cover respectively as a percentage of the total area of forest, cropland and urban area.

**Minor comments**

16-17 - what direct evidence of soil $CO_2$ and dilution is there to support this statement?

Reply: We estimated the impact of those effects by analyzing the temporal pattern of DOC, $CO_2$, and precipitation. For example, DOC and $CO_2$ concentration increased simultaneously from January to April in small rivers (Table 1 and Figure 4 in the revised version of the manuscript), suggesting an increase in terrestrial C inputs. However, an increase in $pCO_2$ was not observed in large rivers during this period even though the DOC increased substantially, which means that

there are processes other than in-stream OC decomposition affecting the $pCO_2$ changes in small rivers. Therefore, soil $CO_2$ input is likely to be responsible for the changes since soil $CO_2$ and in-stream metabolism are the two major sources of riverine $CO_2$ (Yao et al., 2007). Meanwhile, an increase in discharge and decrease in $pCO_2$ has been observed in small rivers from April to July, which indicates that the dilution effect caused by the increasing discharge and depletion effect could be responsible for the decrease. We have rephrased the statement and reduced the certainty. Please refer to Lines 344–355 in the revised version of the manuscript for the changes.

96 - Figure 1 could be supplemented with a land-cover map. Many of the smaller rivers appear to be at higher elevations and I am curious if they are less forested.

Reply: Thank you for your suggestion. We have added a land cover map (Figure 1) in revised version of the manuscript. It appears that the upstream small rivers in the DJRB are more covered with forest compared with the downstream large rivers, and are less impacted by human activities (Figure S3), which together have contributed to the lower $pCO_2$ in small rivers. The impacts of land cover on the spatial pattern of $pCO_2$ have been elaborated in revised manuscript. Please refer to Lines 305–330 in the revised version of the manuscript for the changes.

[Figure]

**Figure S3** Forest, cropland, and urban cover respectively as a percentage of the catchment area. The box mid-lines represent medians; the interquartile range (IQR) is represented by top and bottom of the box, respectively; whiskers indicate the range of 1.5 IQR; the white square symbols represent means, and the other symbols represent Forest, cropland, and urban cover respectively as a percentage of catchment area. Note, this figure has also been added into the Supplementary. Note, this figure has also been added into the Supplementary.

103 - Figure 2's data might be better suited for a bar graph?

Reply: Changed.

[Figure]

**Figure 2** Monthly variations in (a) precipitation of the DJRB and (b) water discharge at the Boluo hydrological gauge, based on data provided by the Hydrological Bureau of Guangdong Province. Note, this figure has also been added into the revised manuscript (Line 105).

163 - I think the reference to equation 2 is incorrect here.

Reply: Changed.

195 - There is no hydrologic data in Table 1. Discharge should be presented.

Reply: Thank you for your advice. Related hydrologic data has been presented in Figure S1 in the supplementary.

[Figure]

**Figure S1** (a) Wet season discharge in Boluo station, based on data provided by the Hydrological Bureau of Guangdong Province. (b) discharge in first order streams

197 - Again there is no stream width or discharge data presented anywhere in the manuscript beside these lines of text.

Reply: Thank you for your advice. Stream width has be presented in Table S1 in the supplementary.

| Stream size | Stream order | Stream width (m) January | April | July | August | October | Wet season | Dry season |
|---|---|---|---|---|---|---|---|---|
| small | | 13.5 ± 9.9 | 16.9 ± 10.1 | 17.1 ± 10.6 | 15.1 ± 9.7 | 14.5 ± 10.9 | 16.3 ± 10.0 | 14.0 ± 10.3 |
| | 1 | 2.9 ± 1.5 | 4.7 ± 2.7 | 6.0 ± 2.1 | 5.6 ± 2.7 | 4.7 ± 2.0 | 5.4 ± 2.4 | 3.8 ± 1.9 |
| | 2 | 8.3 ± 5.2 | 12.7 ± 7.8 | 12.5 ± 7.2 | 10.1 ± 2.8 | 8.4 ±3.2 | 11.8 ± 6.1 | 8.4 ± 4.1 |
| | 3 | 21.1 ± 7.4 | 24.7 ± 5.5 | 24.6 ± 8.5 | 22.1 ± 8.8 | 22.2 ± 10.3 | 23.8 ± 7.6 | 21.7 ± 8.8 |
| large | | 173.9 ± 161.7 | 184.4 ± 164.0 | 187.2 ± 158.5 | 181.1 ± 162.4 | 175.1 ± 165.1 | 184.2 ± 159.1 | 174.5 ± 161.4 |
| | 4 | 67.6 ± 38.5 | 73.0 ±47.1 | 87.0 ±54.7 | 82.3 ±64.4 | 66.2 ± 54.3 | 80.7 ± 54.2 | 66.9 ± 45.8 |
| | 5 | 164.3 ± 46.4 | 187.7 ± 69.6 | 166.6 ±37.6 | 157.0 ±54.9 | 164.6 ± 45.4 | 170.4 ± 53.2 | 164.5 ± 43.3 |
| | 6 | 226.5 ± 23.3 | 235.5 ±37.5 | 241.3 ±45.7 | 231.5 ±44.5 | 243.9 ±38.4 | 236.1 ± 33.4 | 235.2 ± 27.6 |
| | 7 | 425.2 ± 207.3 | 433.4 ± 200.0 | 436.5 ±192.3 | 433.3 ±196.8 | 426.2 ±206.8 | 434.4 ± 177.6 | 425.7 ± 191.7 |

**Table S1** Seasonal variation in stream width for streams from first to seventh order.

202 - U10 is undefined.

Reply: U10 has been defined in the revised version of the manuscript. Now it reads "while wind speed at 1.5 m above the water surface was measured with a Kestrel 2500 handheld anemometer and normalized to a height of 10 m (U10) using the equation from Alin et al. (2011)." (Lines 138–139)
.

275 - DOC and $CO_2$ can simultaneously be transported from terrestrial systems, which also might explain their correlation.

Reply: Because DOC and $CO_2$ can be simultaneously transported from terrestrial systems, changes in DOC concentration could alter OC decomposition but not necessarily leads to the $pCO_2$ pattern. However, the positive relation between river water $pCO_2$ and DOC combined with the negative relation between $pCO_2$ and DO demonstrate that metabolic processes are important for $CO_2$ variation. This is also supported by the inverse relationship between $\Delta CO_2$ and $\Delta O_2$ (Figure 8 in the revised version of the manuscript).

297-318 - This section is very overstated and not the only way to interpret these data. I recommend revising and rephrasing to reduce certainty and include alternative explanations.

Reply: Thank you for your advice. We have rephrased the discussion to reduce the certainty (Please also refer to Lines 293–380 in the revised version of the manuscript), and discussed the potential impact of other factors, including land cover (Lines 305–334 in the revised version of

270 the manuscript), photo-oxidation, and photosynthesis (Lines 374–380 in the revised version of the manuscript).

381-382 - This is possible, but not certain.

275 Reply: Thank you for your advice. We have rephrased the discussion to reduce the certainty. Now it reads

"The difference in seasonal pattern can be explained by the drivers of $pCO_2$ variability as the seasonal variation of riverine $pCO_2$ is likely resulting from the changes of external carbon input,
280 internal production of $CO_2$ (Yao et al., 2007), and the dilution effect caused by precipitation (Johnson et al., 2007). For rivers where $pCO_2$ is lower in summer than in winter, the dilution effect overrides the effect of increased carbon inputs and internal $CO_2$ production (Luo et al., 2019). In contrast, for rivers like the Dongjiang river, although the dilution effect remains, increased $CO_2$ input and metabolism are more significant factors in controlling $pCO_2$, thus
285 leading to higher summer $pCO_2$." Please refer to Lines 425–431 in the revised version of the manuscript for changes.

390 - Respiration and photosynthesis can occur simultaneously.

290 Reply: We agree that respiration and photosynthesis can occur simultaneously. We are interested in the impact of those two processes on $pCO_2$ in our study area. In the nearby Xijiang River (a large river basin in south China with similar climate and landscape as the Dongjiang River basin), high DO and $CO_2$ occurred simultaneously in summer, indicating that photosynthesis is dominant and C source other than respiration should be responsible for high $CO_2$ concentration
295 (Yao et al., 2007). In comparison, DO and riverine $CO_2$ were negatively correlated (Figure 6 in the revised version of the manuscript), and oversaturated $CO_2$ was observed at most sample sites (Figure 8 in the revised version of the manuscript), indicating that the effect of respiration is more obvious. The possible impact of photosynthesis has also been discussed in Lines 124–126 and Lines 376–379 in the revised version of the manuscript.
300

L13. 405 - The units for $pCO_2$ are not consistent (sometimes uatm sometimes ppm). What about Borges 2015, nature geoscience that includes a significant amount of data for rivers in central Africa? Also Mann et al. 2014 JGR-Biogeosciences has additional $pCO_2$ data. Lastly, is the Mississippi River really a subtropical basin?
305

Reply: We apologized for the inconsistent use of units. In some studies, the results of $pCO_2$ were only provided in ppm, and air pressure, an essential for unit transformation, was not provided. Therefore, we have added notes under the table to explain the inconsistent use of units. Thank you for the recommendation. In addition, we have added the data from rivers in Africa(Borges et
310 al., 2015a; Borges et al., 2015b) into the revised version of the manuscript (Table 3, Line 440). For comparison, we try to include data collected under different hydrological conditions. According to the Köppen Climate Classification system, the lower Mississippi river basin belongs to humid subtropical climate zone (Chen and Chen, 2013).

315 L13. 409-412 - Again, I don't think these conclusions are justified

Reply: We have rephrased the sentenses. Now it reads

"Spatial and temporal patterns of $pCO_2$ were mainly affected by terrestrial carbon inputs and in-stream metabolism, both of which varied due to differential catchment settings, land cover, and hydrological conditions."(Lines 455-457 in the revised version of the manuscript)

L13. 417- Still don't really see how depletion would only affect small streams and not the larger ones they flow into?

Reply:. This may result from their different primary controlling process in small and large rivers. $pCO_2$ in small rivers was more affected by soil $CO_2$ input, so as the depletion of soil $CO_2$. For small rivers, the highest value of $pCO_2$ was observed in April (Figure 4), which is consistent with the rapid surge of terrestrial C input, usually occurring at the beginning of the wet season (Hope et al., 2004; Yao et al., 2007; Johnson et al., 2008). However, such an increase in $pCO_2$ was not observed in large rivers (Figure 4), even though DOC in large rivers, increased during the same period, similar to small rivers (Table 1). A possible explanation is that observed $pCO_2$ rise was mainly originated from soil $CO_2$, which was readily emitted from the small rivers into the air, with little reaching the larger rivers downstream (Denfeld et al., 2013; Drake et al., 2018). Differences in $pCO_2$ dynamic in July and August also reflected differential controlling processes in small and large rivers. A decline in $pCO_2$ in July in small rivers suggested that it might have experienced the depletion effect occurring at middle and late wet season (Hope et al., 2004), during which soil $CO_2$ decreased due to the continual precipitation. In contrast, the increase in $pCO_2$ occurring in large rivers in July indicated that the depletion in soil $CO_2$ input could hardly affect the $pCO_2$ in large rivers during this period. Instead, stronger in-stream metabolism caused by OC input and favorable conditions for OC decomposition is more likely to be responsible for the rising $pCO_2$. Please also refer to Lines 344–359 in the revised version of the manuscript for changes.

Referee #2:

My one main concern is in relation to the way the dataset was collected. It appears that the data was not collected with replicates and in a kind of snapshot approach across a large river basin. It is therefore very challenging to standardise for hydrological conditions, time of day etc. when co-ordinating sampling from such a large basin. However, there should be some analyses and discussion around this point to explore how this might impact the results as presented. One of the aims of the study is to "investigate the spatial and temporal pattern of pCO2 and CO2 emission along stream size spectrum" – how would different sampling conditions affect this? Not enough context is provided to reassure the reader that artefacts of the sampling process are not driving at least some of the variability observed in this dataset.

Reply: We thank the reviewer's comments. It is challenging to sample from such a large basin. However, we have put great efforts into reducing the artefacts when designing the sampling strategy (Please refer to Lines 109–124 in the revised version of the manuscript for changes. ). We have carefully chosen the location and time of our fieldwork campaigns. In total, there were 43 sampling sites from seven Strahler stream orders, including 21 large rivers (i.e., fourth

to seven order streams, including mainstem and major tributaries) and 22 small rivers (i.e., first to third order headwater streams). Those sampling sites were widely distributed in the mainstem and nine major sub-basins with different topographic features and land cover (Figure 1 in Line 365 101 of the manuscript and Figure S4 in the supplementary).

In order to investigate $CO_2$ emissions under different hydrological conditions, we performed five fieldwork campaigns from December 2018 to October 2019, including three in the wet season (early wet season - late April, middle wet season - early July, and late wet season - late August) 370 and two in the dry season (middle dry season - December 2018 to early January 2019 and early dry season - late October 2019. Sample sites were measured during the daytime over two weeks inr each field trip. Three rounds of campaigns in the wet season allow each sample site to be measured under different hydrological conditions, and the two-week duration of each campaign allowed streams with different orders and sizes to be measured under various discharges. As for 375 the dry season, the hydrological condition was relatively stable due to low precipitation.

However, there are some artefacts that we were unable to avoid. The possible impacts have been discussed in the revised version of the manuscript (Lines 124–126). For example, our sample sites were measured in the daytime which may lead to underestimate in $pCO_2$ and $CO_2$ emission 380 (Reiman and Xu, 2019). nocturnal $CO_2$ emission rates in rivers could be 27% greater than the daytime rates (Gómez-Gener et al., 2021). We admit that artefacts of sampling have affected our dataset, but they are unlikely to drive the spatial and temporal patterns. Thank you so much for your insightful comments.

385 L55. Please indicate which references refer to which so that the reader can use this as a pointer towards specific studies which observed one or the other pattern.

Reply: Thank you for your advice. References have been changed accordingly. Now it reads "In addition, different rivers in this region may have contrasting trends in $CO_2$ dynamic due to 390 different underlying controlling factors. Some rivers have the highest $CO_2$ efflux in the wet season (Li et al., 2013; Le et al., 2018; Ni et al., 2019), while others have the highest $CO_2$ efflux in the dry season (Luo et al., 2019)". Please refer to Lines 55–56 in the revised version of the manuscript for changes.

395 L96. What type of forest? Just to clarify, these "plains and hills" are predominantly covered by this forest? Please provide some more information on the extent of coverage.

Reply: The dominant land use of the catchment is highly diverse evergreen forests of broad-leaved and needle-leaved species s (Ran et al., 2012; Chen et al., 2013) , accounting for about 400 64% of the river basin area. Please refer to Lines 94–96 in the revised version of the manuscript for changes.

L127. Please provide details of the flow meter, including accuracy etc.

405 Reply: Flow velocity was determined using a Global Water Flow Probe FP111 with a precision of 0.1 m s$^{-1}$. Please refer to Line 137 in the revised version of the manuscript.

L130. Can you provide an indication of how big this underestimation might be?

Reply: Flow velocity measured near the bank could be about 40% of the maximum flow velocity at the crosssection (Moramarco et al., 2004; Le Coz et al., 2008). Please refer to Lines 140–142 in the revised version of the manuscript.

L142-3. These volumes are larger than what are typically used for headspace extractions. Did you test this method for accuracy compared to smaller volume methods or can you provide a reference to back up this approach? Mostly to confirm that full equilibration between water and headspace is occurring within 1 min of shaking.

Reply: Thank you for your advice. A comparative analysis of the small and large volume headspace method has been conducted to evaluate the accuracy of the headspace extraction method used in this study (Supplementary text1). Now it reads

"Both large volume bottle and small volume syringe were used in headspace equilibration method (Yoon et al., 2016). In this study, a 625 mL reagent bottle and a 100 ml syringe equilibrator were chosen for headspace extraction to investigate the impact of headspace volume and shaking time on the result of $pCO_2$ measurement. For the bottle equilibrator, 400 ml of sample water was collected with 225 ml ambient air, while 50 ml of sample water was collected with 50 ml of ambient air for the syringe equilibrator. According to Hope et al. (1995), equilibrium in the headspace could be achieved after 1 min of vigorous shaking when adapting the rapid headspace analysis technique developed by Kling et al. (1991). However, 5–10 min were also used for $pCO_2$ determination (Abril et al., 2015). Therefore, we measured the $pCO_2$ value in the headspace after shaking the equilibrator for 1–5 min at the one-minute interval. The average of triple replicates was then calculated for comparative analysis (Table S2). The experiment showed that syringe and bottle equilibrator gave very consistent results. Overall, the average $pCO_2$ value of syringe headspace was 1.2% larger than that of the bottle headspace, which was less than the 1.5% error of Li-850. However, it could take more time for a large volume equilibrator to achieve equilibrium. The $pCO_2$ results of the small volume syringe headspace are not significantly different after more than two minutes of shaking, while it took the large volume bottle equilibrator three minutes to achieve a similar $pCO_2$ value. Therefore, for rapid field measurement of surface water $pCO_2$, shaking for two minutes when using syringe headspace and three minutes when using the bottle headspace could yield reliable test results (Figure S3). In this study, we vigorously shake the bottle equilibrator for at least 1 minute (usually 1-3 minutes), which might cause a 1–5% underestimate of the $pCO_2$ result. Furthermore, Koschorreck et al. (2021) found that reducing the headspace ratio could significantly increase the accuracy of the headspace method. Therefore, a large volume equilibrator might be more suitable for the field measurement since a low headspace ratio could be easier to achieve.

"

**Table S2** Comparison of measured $pCO_2$ using two headspace extraction methods.

| Equilibrator | Shaking time |
| --- | --- |

|  |  | 1 min | 2 min | 3 min | 4 min | 5 min |
|---|---|---|---|---|---|---|
| Bottle | test 1 | 637.9 | 642.3 | 669.1 | 664.6 | 673.1 |
|  | tes 2 | 645.3 | 651.7 | 658.2 | 666.5 | 680.0 |
|  | tese 3 | 634.4 | 645.8 | 665.2 | 662.1 | 664.6 |
|  | Average | 639.2 | 646.6 | 664.2 | 664.4 | 672.6 |
| Syringe | tes 1 | 640.6 | 662.7 | 664.9 | 681.5 | 681.5 |
|  | tes 2 | 639.5 | 670.4 | 670.4 | 674.8 | 680.3 |
|  | tes 3 | 648.4 | 660.5 | 664.9 | 669.3 | 666.0 |
|  | Average | 642.8 | 664.5 | 666.7 | 675.2 | 675.9 |

Note, this table has also been added into the Supplementary.

[Figure]

**Figure S2** Measured $pCO_2$ in headspace after shaking for 1-5 minutes. Note, this figure has also been added into the Supplementary.

L162. Think this is supposed to be eq 3.

Reply: Changed.

L189. Does this include replicates at any sites? Or were single measurements only of $FCO_2$ and pCO2 undertaken at each site? It seems strange to omit any kind of replication at each

measurement site, so I would encourage the authors to explain why and discuss whether this lack of replication had any major impact on their findings.

465  Further, were sites measured all in the same day or over multiple days? If so, how might time of day or hydrologic conditions varied across these measurements within each campaign? I know you can't go back and fix any of these potential issues after the fact, but some discussion of potential issues here would be useful to convince the reader that there these decisions made when designing the sampling strategy have not substantially impacted the data that is presented here.

470 This is most concerning when I look at Table 1. The values appear very consistent across all the sites, yet the standard deviation compared to the means are very large in some cases.

Reply: The measurements of $pCO_2$ at each site were repeated twice, and the average was then calculated to represent the surface $pCO_2$. The variation between the two measurements is less than 5%, and the accuracy of Li-850 is within 1.5% of the reading (Lines 158–160 in the revised version of the manuscript). However, for most sites, only one successful measurement of $FCO_2$ was conducted. A successful chamber measurement could take several deployments because chamber could encounter obstacles or got stranded on underwater banks, especially in small rivers. Therefore chamber was deployed for multiple times until it can drift freely without being stopped by obstacles. We do conduct more than one successful meansurement in several sites and the results were consistent. The drifting allowed chamber to measure the $FCO_2$ along the reach, which could reduce the possible bias of conducting measurement at a single location. We understand that this is not an ideal sampling strategy, but it is unlikely having substantial impacts on the result.Considering that the chamber measurement is mainly used for the comparison of k600 and $FCO_2$ between dry and wet season and between small and large rivers, five measurements at each site (three in wet season and two in dry season) could still reflect $CO_2$ emission under different hydrological conditions in streams with different sizes. For small rivers, 33 and 52 measurements were conducted in dry and wet season, respectively. For large river, 42 and 61 measurements were conducted in dry and wet season, respectively.

490

In order to investigate $CO_2$ emissions under different hydrological conditions, we performed five fieldwork campaigns from December 2018 to October 2019, including three in the wet season (early wet season - late April, middle wet season - early July, and late wet season - late August) and two in the dry season (middle dry season - December 2018 to early January 2019 and early dry season - late October 2019. Sample sites were measured in the daytime over two weeks for each field trip. Three rounds of campaigns in the wet season allow each sample site to be measured under different hydrological conditions, and the two-week duration of each campaign allowed streams with different orders and sizes to be measured under various discharges. As for the dry season, the hydrological condition was relatively stable due to low precipitation. However, field measurements conducted during the daytime could lead to underestimate in $pCO_2$ and $CO_2$ emission (Reiman and Xu, 2019). nocturnal $CO_2$ emission rates in rivers could be 27% greater than the daytime rates (Gómez-Gener et al., 2021).

L197. Change Q to "discharge"

505

Reply: Changed.

L225. What did this "strongest increase" actually relate to? Stream order is just a proxy for many things, including discharge, catchment characteristics etc. This is not fully discussed or addressed in the discussion.

Reply: Initially, we expected a gradual decrease or increase in $pCO_2$ from low order to high order streams. However, we only observed a strong increase in $pCO_2$ from the third order stream to the fourth order stream, it was stronger than any other two consecutive stream orders. Meanwhile, the first to third order streams have similar $pCO_2$ values, and the fourth to seventh order streams have similar $pCO_2$ values. As such, we discussed the underlying mechanisms that regulate the differences in the $pCO_2$ between small rivers (the first to third order streams) and large rivers (the fourth to seventh order streams).

L245. Clarify the sentence here: "indicating that the majority of the river network is a carbon source".

Reply: Changed.

L259. Not sure what this means, between wet vs dry seasons?

Reply: Thank you for your comment. Among five fieldwork campaigns, two of them were performed in dry season and three of them were performed in wet season. Here we compare the result between different campaigns in the same hydrological season. We have clarified it in the revised version of the manuscript. now it reads
"However, comparisons between different phases in the same hydrological period (e.g., early, middle, and later wet season) did not differ significantly (paired sample t test, p > 0.05) for both river size classes." (Lines 278–280)

L264. High compared to what?

Reply: It is higher than other months.

L273. This too broad a statement to really be useful. This is dependent on the river and its setting etc. Perhaps rethink the purpose of this opening sentence and target it more directly to the immediate discussion.

Reply: Thank you so much for your advice. Based on your suggestion, we have reworded the statement to make it more specific and relevant to this study. Now it reads '"The spatial pattern of $pCO_2$ in the DJRB is likely resulting from changes in the intensity of in-stream metabolism.". Then, we immediately started the discussion about how in-stream metabolism could differ in small and large rivers. Please also refer to Lines 293–294 in the revised version of the manuscript for the changes.

L310. Which "should" lead to a decrease? Because you then observed $pCO_2$ to increase, rather than be diluted.

Reply: In small rivers, a decrease in $pCO_2$ in July was observed, and it was likely the result of the $CO_2$ depletion effect in soils combined with the dilution effect of precipitation. The dilution effect leads to a decrease in $pCO_2$. We have rephrased it in the revised version of the manuscript (Lines 353-355) Now it reads

"A decline in $pCO_2$ in July in small rivers suggested that it might have experienced the depletion effect occurring at middle and late wet season (Hope et al., 2004), during which soil $CO_2$ decreased due to the continual precipitation."

RC: L322. Decomposition of organic carbon "within the water column" (internal DOC decomposition)?

Reply: Changed.

L331. Plenty of studies have indicated that DOC can be readily decomposed in headwater streams, e.g. Vonk et al. 2013 (doi: 10.1002/grl.50348), Dean et al. 2019 (doi:10.1029/2018JG004650).

Reply: Thank you for your recommendation. Indeed, DOC can be readily decomposed in some headwater streams, but it also depends on their setting. Headwater streams in peatland or permafrost regions (Vonk et al., 2013; Dean et al., 2019) tend to have low gradients and more favorable conditions for DOC decomposition. In contrast, the headwater streams in the Dongjiang River basin usually have high channel gradient and high flow velocity due to a predominantly hilly landscape. Therefore, it will be more difficult for DOC to be decomposed here.

L334. Should you not then see a correlation between DOC and $pCO_2$?

Reply: We have observed a positive relation between river water $pCO_2$ and DOC (Figure 6b in the revised version of manucscript). Moreover, the discrepancy in seasonal changes of DOC and $pCO_2$ was also observed in our data. Increased DOC in large rivers from January to April did not lead to the increase in $pCO_2$. A possible explanation for such phenomenon is that the effect of increasing DOC on $pCO_2$ is confined by other factors that control the intensity of in-stream metabolism. For large rivers, relatively low temperature and short water residence time due to high flow velocity may have led to the low $pCO_2$ in April despite increased DOC. Therefore, even though the higher $pCO_2$ in large rivers relative to small rivers was associated with stronger in-stream metabolism, it might not be controlled by DOC concentration.

L343. In line with previous studies, e.g. Long et al. 2015 (doi: 10.1002/2015JG002955). Fig 8. I suggest repositioning the legend so that single blue dot is more obvious.

Reply: Thank you for your advice. The legend has been repositioned.

[Figure]

**Figure 9** Relationship between $k_{600}$ and flow velocity. The dashed line represents the parameterization of (Alin et al., 2011).

L393. For all rivers? Or large rivers? Because the earlier discussion suggested internal production of $CO_2$ was more important for the larger rivers.

Reply: A negative relationship between DO and $pCO_2$ has been observed in both small and large rivers (Figure 6 in the revised manuscript), which suggests that internal production of $CO_2$ occurs in both of them. Undersaturation of DO and supersaturation of $pCO_2$ have been observed in both small and large rivers, which means that internal production of $CO_2$ occurs in both small and large rivers, even though in-stream metabolism was more important for the large rivers (Figure 6 in the revised manuscript). In contrast, supersaturation DO and supersaturation $CO_2$ occurred simultaneously in summer in the nearby Xijiang river (Yao et al., 2007), indicating that photosynthesis is stronger than respiration and other C sources should be responsible for high $pCO_2$ concentration. Therefore, compared with Xijaing River, the contribution of internal production of $CO_2$ on $pCO_2$ is more obvious for both small and large rivers in the Dongjiang River Basin. Even though small rivers were also under the influence of lateral soil $CO_2$ input.

---

## Author Response (AR2)

Dear Dr. Bouillon,

Thank you very much for the thoughtful review of our manuscript. Based on your and the reviewer's comments, we have further improved the manuscript. Please find below our point-by-point response (in bright blue) to the reviewer's comments. All changes have also been highlighted in yellow in the track change file of the manuscript. The line numbers refer to the lines in the revised version.

We hope that the revised manuscript will be acceptable for publication in Biogeosciences.

Thank you very much for your kind consideration.

With best regards
Lishan Ran, on behalf of all co-authors
* * *
**Primary criticisms**

The authors present a revised manuscript that now includes some analyses of the differences in river C by land-cover, which are welcomed. However, it is my view that the manuscript is still overly specific and confident about the origins and processes controlling $CO_2$ concentrations ($pCO_2$) when they do not have the data to rule out other explanations. Specifically, the authors still suggest that the spatial patterns observed are due primarily to in-stream metabolism.
Reply: Thank you very much for your comments. Based on your comments and suggestions, we have reframed the discussion on the drivers of the spatial pattern of the stream water $pCO_2$. Instead of focusing on the dominant role of in-stream metabolism, we now discuss how land-cover and catchment topography have affected the spatial pattern by influencing terrestrial carbon inputs and in-stream metabolism. Furthermore, based on your suggestions, we have reduced the certainty and discussed the potential impacts of other factors, including the higher soil respiration in cropland-impacted large river catchments, high gas exchange velocity in small rivers, and carbonate buffering. Please refer to Lines 293-361 in the revised version of the manuscript for the changes.

The land-use data they now present however, shows that the big and small rivers diverge in terms of %cropland/urban, with the larger rivers exhibiting higher proportions of impacted areas. The authors are correct that this could lead to higher inputs of labile DOC, but they do not provide evidence of this process. Further, higher soil respiration in these impacted zones could also generate higher soil $CO_2$, which is subsequently transported to the river.
Reply: Thank you very much for your suggestions. We have revised the discussion on the impact of land use on riverine $pCO_2$ by analyzing two processes that control the amount and lability of carbon transported from cropland to rivers. On one hand, cropland could provide a more favorable condition for soil erosion and the transfer of terrestrial carbon from land to rivers, contributing to a higher $pCO_2$. On the other hand, intensification of agricultural practices could promote the decomposition of soil organic matter, thereby increasing the concentration of $CO_2$ and liable DOC in the soil (Borges et al., 2018). The soil $CO_2$ could be easily transported to rivers, while the liable DOC component could be decomposed rapidly after entering the rivers due to their sensitivity to in-stream metabolism (Lambert et al., 2017; Li et al., 2019). Therefore, we have discussed the possible impacts of both processes in explaining the high $pCO_2$ in cropland-impacted rivers in the revised manuscript and cited the references to support our arguments. Please refer to Lines 297-301 in the revised version of the manuscript for the changes.

In addition to land-use differences, the authors measured differences in k600 (gas transfer velocities) between the large and small rivers. It is well known that k600 and $pCO_2$ vary inversely (Rocher-Ros 2019, LOL), which also might explain the elevated $pCO_2$ concentrations in the larger rivers. In other words, in-stream metabolism might be very similar between rivers/streams of all sizes, but simply the outgassing is higher in smaller and more turbulent streams, resulting in lower $pCO_2$. This seems in line with the similar $pCO_2/O_2$ trends the authors observed between the two size groups.
Reply: Thank you very much for your comment. We agree that the high $k_{600}$ in small rivers could contribute to their relatively low $pCO_2$. Due to steeper slopes and higher flow velocities, small rivers in the DJRB tend to have higher $k_{600}$. As a consequence, $CO_2$ in small rivers can emit into the atmosphere more rapidly, preventing the build-up of dissolved $CO_2$ and thus lower $pCO_2$. Therefore, based on your suggestions, we have discussed the impact of $CO_2$ emissions on riverine $pCO_2$ in small rivers. Please refer to Lines 317-321 in the revised manuscript for the changes. However, small rivers in the DJRB are much less turbulent than highly turbulent streams (i.e., $k_{600} > 100$ m d$^{-1}$) as reported by Rocher-Ros et al. (2019) . The mean $k_{600}$ values in small rivers of the DJRB were only $8.29 \pm 11.29$ m d$^{-1}$ and $4.90 \pm 3.82$ m d$^{-1}$ for the wet season and the dry season, respectively. Therefore, it is unlikely that the spatial pattern was primarily controlled by the outgassing of $CO_2$ from streams. Additional processes have facilitated the carbon transfer from small rivers to downstream large rivers, supporting the higher $pCO_2$ in large rivers.

Indeed, we have observed a pronounced presence of in-stream metabolism in both small and large rivers. However, the difference in the $\Delta CO_2:\Delta O_2$ stoichiometry between small and large rivers suggested the different strength of in-stream metabolism (Rasera et al., 2013). The

$\Delta CO_2:\Delta O_2$ stoichiometry in large rivers is closer to the 1:1 line than that in small rivers, indicating that large rivers are more affected by the metabolic processes (Jeffrey et al., 2018; Amaral et al., 2020). For large rivers, the linear regression is $\Delta CO_2 = -0.999 (\pm 0.081) \Delta O_2$ $+18.020 (\pm 5.995)$ ($r^2 = 0.62$, $p < 0.001$). When the $CO_2$ concentration increases in large rivers, a similar magnitude of decrease in dissolved $O_2$ concentration occurs, indicating that in-stream metabolism is the primary control on $pCO_2$. In contrast, the linear regression for small rivers is $\Delta CO_2 = -0.868 (\pm 0.098) \Delta O_2 + 21.42 (\pm 4.175)$ ($r^2 = 0.41$, $p < 0.001$), which means that with the $CO_2$ concentration increasing by 1 µmol L$^{-1}$, the $O_2$ concentration decreases by only 0.868 µmol L$^{-1}$. Therefore, extra $CO_2$ inputs have contributed to the changes in $pCO_2$ despite the strong presence of in-stream metabolism. We have revised the manuscript by discussing the similarity and differences between the two size groups regarding in-stream metabolism. Please refer to Lines 341-361 in the revised manuscript for the changes.

Lastly, alkalinity has been shown to buffer and create $CO_2$ over saturation in natural waters (Stets 2017). Based on the aggregated table 1, it seems like the larger rivers do indeed have
higher alkalinity.

Reply: Thank you very much for your comment. We agree that alkalinity could buffer and create $CO_2$ oversaturation in natural waters. Carbonate buffering could decrease the $CO_2$ emissions in small rivers by increasing the ionization of $CO_2$, resulting in increased transfer of DIC and higher $pCO_2$ in downstream large rivers (Stets et al., 2017). However, strong carbonate buffering
usually occurs in high-alkalinity ($>2500$ $\mu mol$ $L^{-1}$) streams with high pH ($>8$), while in low-alkalinity waters, the pool of ionized $CO_2$ is relatively small, indicating a weak carbonate buffering (Stets et al., 2017). Since the streams in the DJRB are characterized by low alkalinity ($726 \pm 364$ $\mu mol$ $L^{-1}$ and $844 \pm 409$ $\mu mol$ $L^{-1}$ for small and large rivers, respectively), carbonate buffering is unlikely a major contributor to the high $pCO_2$ in large rivers, even though slightly
higher alkalinity has been observed in large rivers. We have discussed the possible impacts of carbonate buffering. Please refer to Lines 321-328 in the revised manuscript for the changes.

Ultimately, I think that some reframing is still needed, and the authors should be less certain with their interpretations around metabolism as the primary control given that a plethora of additional
controls (listed above) could also affect $pCO_2$. Moreover, I'd recommend perhaps abandoning the artificially divide between small/large rivers and just use discharge to examine effects of river size.

Reply: Thank you very much for your comments and suggestions. We have revised the discussion on the drivers of the spatial and temporal patterns. We started with analyzing the
impacts of land cover and catchment topography on the spatial pattern of $pCO_2$, then the temporal pattern and its responses to precipitation and temperature seasonality, before finishing on other minor controlling factors. Based on the referee's comment, we have further examined the impact of other factors, including $k_{600}$, carbonate buffering, and increased soil respiration in cropland. The interpretations around metabolism were also revised. Please refer to Lines 293-389
in the revised manuscript for the changes.

We fully agree with the reviewer that discharge could greatly alter $pCO_2$ and $CO_2$ emissions and is an important hydrological attribute to examine the effects of river size on stream water $pCO_2$ and $CO_2$ emissions. For the DJRB with a clear seasonal pattern in flow discharge, however, we
noted that one river could be divided into different size groups in different seasons according to its discharge size. This may affect our discussion on the $pCO_2$ and $CO_2$ emission difference between small and large rivers. In addition, stream order has been commonly used as a parameter when upscaling $CO_2$ emissions from regional and global river networks (Butman and Raymond, 2011; Raymond et al., 2013; Marescaux et al., 2018). Consequently, the spatial and temporal
distribution of $pCO_2$ and $CO_2$ emissions along the stream size spectrum have been widely used to estimate regional and global $CO_2$ emission flux. In this manuscript, we tend to retain the stream size based on the Strahler stream orders, but we will also consider the effect of discharge on $pCO_2$ and $CO_2$ emissions in our future studies. We are very grateful for your constructive and useful comments.

**Specific comments**

- Incorrect usage of "hinges"

Reply: Thank you for the comment. We have replaced the "hinges" with "prohibits".

- I still don't see the evidence for this claim in the paper
Reply: We have revised the discussion on the drivers of the spatial and temporal patterns. We started with analyzing the impacts of land cover and catchment topography on the spatial pattern of $pCO_2$, then the temporal pattern and its response to precipitation and temperature seasonality.

Please refer to Lines 292-388 in the revised manuscript for the changes.

24-25 - Is the lack of difference the total magnitude or the areal flux? Very different implications…
Reply: We apologize for the confusing statements. Small and large rivers have similar areal $CO_2$

fluxes. Small rivers have a higher gas transfer velocity ($k$) and lower $pCO_2$, while large rivers have a lower $k$ value and higher $pCO_2$. We have further clarified this in the revised version of the manuscript, please refer to Line 24 for the change.

56-57 - Is it necessarily runoff or could it be other seasonal factors (temp/plant seasonality/etc.)

Reply: Thank you very much for the comment. The rivers mentioned here are all located in the subtropical monsoon climate zone and have similar temperature and plant seasonality. The wet season has higher temperature and net primary productivity. Other factors may have affected the seasonal changes of $pCO_2$, but it is more likely that the increase of runoff during the wet season has contributed to the two distinct patterns of the $pCO_2$ dynamics. On one hand, recent studies have showed that the increased runoff could enhance external carbon inputs and thus $CO_2$ emissions in some rivers (Hope et al., 2004; Johnson et al., 2008). On the other hand, the increased runoff may result in a dilution of the dissolved $CO_2$ concentration in river waters (Ran et al., 2017; Li et al., 2018). We have further clarified this in the revised version of the manuscript, please refer to Lines 56-62 for the change.

81-82 - I'd stick with either evasion or emission, for consistency
Reply: Thank you for your comment. We have replaced "evasion" with "emission" throughout the text.

Figure 1 - The land-use areas do not make sense as currently displayed. Since this is one entire basin, the MDJRB should include the UDJRB areas and the LDJRB should include both since all upstream water flows into these lower parts of the basin… ultimately, the upstream land-cover should be calculated for each sampling point using the sub-catchment outlines.
Reply: Thank you for your comment. The land-use area displayed in Figure 1 is mainly used to show the difference in land use from upstream to downstream. Considering most of our sample sites are located in tributaries, they are not under the influence of land use in the upper part of the basin. Thus, it may not be necessary to include the land cover in upper regions. For each sampling point, the upstream land cover has been calculated using the sub-catchment outlines as recommended by the referee.

- Headspace is misspelled in the equation
Reply: Changed. Thank you very much for pointing this out.

- Do you mean decomposition here?

Reply: Yes, in the DJR, decomposition of OC is the primary form of in-stream metabolism. We have clarified this in the revised version of the manuscript. Please refer to Lines 383-388 for the changes.

    297 - I don't think this is necessarily true! Many studies show higher rates of metabolism in small streams, which receive higher proportions of labile material from their proximity to recent terrestrial inputs.

    Reply: Thank you very much for your comment. Indeed, DOC can be readily decomposed in some headwater streams, but it also depends on their setting. For example, the headwater streams in peatland or permafrost regions (Vonk et al., 2013; Dean et al., 2019) tend to have low gradients and more favourable conditions for DOC decomposition. In contrast, the headwater streams in the Dongjiang River basin usually have steep channel slopes and high flow velocities due to a predominantly hilly landscape. Therefore, it would be more difficult for DOC to be decomposed here. We have revised the discussion about OC decomposition in small rivers. Please refer to Lines 335-337 in the revised manuscript for the changes.

    Also, I would use "Terrestrial organic carbon is" rather than the plural form.

    Reply: Changed.

    306 - I don't think it's appropriate to cite this reference in support of trends you are describing in your own study.

    Reply: Thank you for your comment. We have removed the reference from the text.

    307 - Here you cite Figure S3 but I think it should be Fig 7?

    Reply: Changed. Thank you very much for pointing this out.

    366- Should this be Figure 8?

    Reply:  Changed. Thank you very much for pointing this out.

    371-374 - Are the y-intercepts statistically different? They are very close regardless…

Reply: Thank you very much for your comment. The p-value of the y-intercepts is 0.048, so they are statistically different. In order to clarify the differences between small and large rivers, we have substantially revised the discussion on the strength of in-stream metabolism in small and large rivers. Please refer to Lines 343-355 in the revised manuscript for the changes.

375- Abrupt transition after being so certain that metabolic processes govern $p$CO$_2$…

    Reply: Thank you very much for your comment. Here we discussed why other factors are unlikely to be the primary process that govern the $p$CO$_2$ dynamics, which is consistent with our previous discussion about why the metabolic process is important for the $p$CO$_2$ dynamics.

377 - You do not have diel measurements so how do you know the effect of photosynthesis? Perhaps DO drops much lower at night?

    Reply: Thank you very much for the comment. As the reviewer noted, because we did not conduct diel measurements, we cannot calculate the rates of aquatic photosynthesis and respiration in this manuscript. However, we can compare the effect of photosynthesis and in- stream metabolism based on the concentrations of DO and dissolved CO$_2$. The unsaturated DO

indicates that the overall rate of respiration is higher than that of photosynthesis. Therefore, even if the DO drops much lower at night, it is unlikely that the rate of photosynthesis could be overwhelmingly higher than that of respiration at the daytime.

**References**

Amaral, J. H. F., Melack, J. M., Barbosa, P. M., MacIntyre, S., Kasper, D., Cortés, A., Silva, T. S. F., Nunes de Sousa, R., and Forsberg, B. R.: Carbon dioxide fluxes to the atmosphere from waters within flooded forests in the Amazon basin, Journal of Geophysical Research: Biogeosciences, 125,
e2019JG005293, https://doi.org/10.1029/2019JG005293, 2020.

Borges, A. V., Darchambeau, F., Lambert, T., Bouillon, S., Morana, C., Brouyere, S., Hakoun, V., Jurado, A., Tseng, H. C., Descy, J. P., and Roland, F. A. E.: Effects of agricultural land use on fluvial carbon dioxide, methane and nitrous oxide concentrations in a large European river, the Meuse (Belgium), Science of The Total Environment, 610-611, 342-355,
https://doi.org/10.1016/j.scitotenv.2017.08.047, 2018.

Butman, D., and Raymond, P. A.: Significant efflux of carbon dioxide from streams and rivers in the United States, Nature Geoscience, 4, 839-842, 2011.

Dean, J. F., Garnett, M. H., Spyrakos, E., and Billett, M. F.: The potential hidden age of dissolved organic carbon exported by peatland streams, Journal of Geophysical Research: Biogeosciences, 124, 328-341,
2019.

Hope, D., Palmer, S. M., Billett, M. F., and Dawson, J. J. J. H. P.: Variations in dissolved $CO_2$ and $CH_4$ in a first-order stream and catchment: an investigation of soil–stream linkages, Journal of Hydrological Processes, 18, 3255-3275, https://doi.org/10.1002/hyp.5657, 2004.

Jeffrey, L. C., Santos, I. R., Tait, D. R., Makings, U., and Maher, D. T.: Seasonal drivers of carbon
dioxide dynamics in a hydrologically modified subtropical tidal river and estuary (Caboolture River, Australia), Journal of Geophysical Research: Biogeosciences, 123, 1827-1849, https://doi.org/10.1029/2017jg004023, 2018.

Johnson, M. S., Lehmann, J., Riha, S. J., Krusche, A. V., Richey, J. E., Ometto, J. P. H., and Couto, E. G.: $CO_2$ efflux from Amazonian headwater streams represents a significant fate for deep soil respiration,
Geophysical Research Letters, 35, https://doi.org/10.1029/2008GL034619, 2008.

Lambert, T., Bouillon, S., Darchambeau, F., Morana, C., Roland, F. A. E., Descy, J.-P., and Borges, A. V.: Effects of human land use on the terrestrial and aquatic sources of fluvial organic matter in a temperate river basin (The Meuse River, Belgium), Biogeochemistry, 136, 191-211, 10.1007/s10533-017-0387-9, 2017.

Li, S., Ni, M., Mao, R., and Bush, R. T.: Riverine $CO_2$ supersaturation and outgassing in a subtropical monsoonal mountainous area (Three Gorges Reservoir Region) of China, Journal of Hydrology, 558, 460-469, https://doi.org/10.1016/j.jhydrol.2018.01.057, 2018.

Li, X., Xu, J., Shi, Z., and Li, R.: Response of Bacterial Metabolic Activity to the River Discharge in the Pearl River Estuary: Implication for CO2 Degassing Fluxes, Frontiers in Microbiology, 10,
10.3389/fmicb.2019.01026, 2019.

Marescaux, A., Thieu, V., and Garnier, J.: Carbon dioxide, methane and nitrous oxide emissions from the human-impacted Seine watershed in France, Science of the Total Environment, 643, 247-259, 2018.

Ran, L., Lu, X. X., and Liu, S.: Dynamics of riverine $CO_2$ in the Yangtze River fluvial network and their implications for carbon evasion, Biogeosciences, 14, 2183-2198, https://doi.org/10.5194/bg-14-2183-
2017, 2017.

Rasera, M. d. F. F., Krusche, A. V., Richey, J. E., Ballester, M. V., and Victoria, R. L.: Spatial and temporal variability of $p$$CO_2$ and $CO_2$ efflux in seven Amazonian Rivers, Biogeochemistry, 116, 241-259, https://doi.org/10.1007/s10533-013-9854-0, 2013.

Raymond, P. A., Hartmann, J., Lauerwald, R., Sobek, S., McDonald, C., Hoover, M., Butman, D., Striegl, R., Mayorga, E., and Humborg, C.: Global carbon dioxide emissions from inland waters, Nature, 503, 355-359, https://doi.org/10.1038/nature12760, 2013.

Rocher-Ros, G., Sponseller, R. A., Lidberg, W., Mörth, C. M., and Giesler, R.: Landscape process domains drive patterns of CO2 evasion from river networks, Limnology and Oceanography Letters, 4, 87-95, 2019.

Stets, E. G., Butman, D., McDonald, C. P., Stackpoole, S. M., DeGrandpre, M. D., and Striegl, R. G.: Carbonate buffering and metabolic controls on carbon dioxide in rivers, Global Biogeochemical Cycles, 31, 663-677, https://doi.org/10.1002/2016gb005578, 2017.

Vonk, J. E., Mann, P. J., Davydov, S., Davydova, A., Spencer, R. G., Schade, J., Sobczak, W. V., Zimov, N., Zimov, S., and Bulygina, E.: High biolability of ancient permafrost carbon upon thaw, Geophysical
Research Letters, 40, 2689-2693, 2013.

---

## Author Response (AR3)

Dear Dr. Bouillon,

Thank you very much for the thoughtful review of our manuscript. Based on your comments, we have further improved the manuscript. Please find below our point-by-point response (in bright blue) to your comments. All changes have also been highlighted in yellow in the track-changes file. The line numbers refer to the lines in the revised version.

We hope that the revised manuscript is now acceptable for publication in Biogeosciences.

Thank you very much for your kind consideration.

With best regards
Lishan Ran, on behalf of all co-authors
* * *
L13: emission measurement : emission measurements
Reply: Revised.

L16-18: this new sentence is intended to address one of the key issues raised by Ref #1, i.e. that $pCO_2$ is governed not just by in situ metabolism. While this sentence captures that, it is not fully clear unless you specifically mention that 'terrestrial carbon inputs' also include inorganic C (i.e. $CO_2$). The same holds for other versions of this statement on L 292-293 and L464-465.
Reply: Thank you for your comments and suggestions. We agreed with your comments. Based on your suggestions, we have rephrased these statements in the revised version. For Line16–18, now it reads "Spatial and temporal patterns of $pCO_2$ were mainly affected by terrestrial carbon inputs (i.e., organic and inorganic carbon) and in-stream metabolism, both of which varied due to different land cover, catchment topography, and seasonality of precipitation and temperature." Please also refer to Lines 16–18, Lines 292–293, and Lines 462–465 in the revised version of the manuscript for the changes.

L108: Field measurment and analysis: Field measurements and analyses
Reply: Revised.

L318: CO2 in small rivers can emit into : $CO_2$ in small rivers can exchange with
Reply: Revised.

L323: "and thus the higher $p$CO$_2$ in downstream large rivers": awkward, rephrase.
Reply: Thank you very much for your comments and suggestions. We have rephrased this sentence. Now it reads "Recent studies indicate that carbonate buffering could decrease the CO$_2$ emissions from small rivers by increasing the ionization of CO$_2$ (Stets et al., 2017), thereby increasing the transfer of DIC towards the rivers downstream, which resulted in the higher $p$CO$_2$ in downstream large rivers." Please also refer to Lines 323–325 in the revised version of the manuscript for the changes.

L335-337: please remove this or reformulate. I do not see the logic of invoking the absence of anoxic environments to explain this.
Reply: Thank you for your comments and suggestions. We have removed the sentence from the manuscript.

L367: "This could either enhance directly riverine $p$CO$_2$ or fuel OC decomposition": This could either directly increase riverine $p$CO$_2$, or fuel OC decomposition.
Reply: Revised. Again, thank you very much for your constructive comments, which have greatly improved the manuscript.